

# Stratospheric ozone trends for 1984–2021 in the SAGE II – OSIRIS – SAGE III/ISS composite dataset

Kristof Bognar[1], Susann Tegtmeier[1], Adam Bourassa[1], Chris Roth[1,*], Taran Warnock[1], Daniel Zawada[1], and Doug Degenstein[1]

[1]Institute of Space and Atmospheric Studies, University of Saskatchewan, Saskatoon, SK, Canada
[*]Now at Vehicle Prognostics Research Group, Hamilton, ON, Canada

**Correspondence:** Kristof Bognar (christopher.bognar@gmail.com), Doug Degenstein (doug.degenstein@usask.ca)

**Abstract.**

After decades of depletion in the $20^{th}$ century, near-global ozone now shows clear signs of recovery in the upper stratosphere. The ozone column, however, remained largely constant since the turn of the century, mainly due to changes in the lower stratosphere. In the tropical lower stratosphere, ozone is expected to decrease as a consequence of enhanced upwelling

driven by increasing greenhouse gas concentrations, and this is consistent with observations. There is recent evidence, however, that mid-latitude ozone continues to decrease as well, contrary to model predictions. These changes are likely related to dynamical variability, but the impact of changing circulation patterns on stratospheric ozone is not well understood. Here we use merged measurements from the Stratospheric Aerosol and Gas Experiment II (SAGE II), the Optical Spectrograph and InfraRed Imaging System (OSIRIS), and SAGE III/ISS to quantify ozone changes in the 2000–2021 period. We implement

a sampling correction for the OSIRIS and SAGE III/ISS datasets, and assess trend significance taking into account temporal differences with respect to Aura Microwave Limb Sounder data. We show that ozone increased by 2–6 % in the upper and 1–3 % in the middle stratosphere since 2000, while lower stratospheric ozone decreased by similar amounts. These decreases are significant in the tropics (>95 % confidence), but not necessarily at mid-latitudes (>80 % confidence). In the upper and middle stratosphere, changes since 2010 point to hemispheric asymmetries in ozone recovery. Significant positive trends are

present in the southern hemisphere, while ozone at northern mid-latitudes has remained largely unchanged in the last decade. These differences might be related to asymmetries and long-term variability in the Brewer-Dobson circulation. Circulation changes impact ozone in the lower stratosphere as well. In tropopause relative coordinates, most of the negative trends in the tropics lose significance, highlighting the impacts of a warming tropopause and increasing tropopause altitudes.

## 1   Introduction

Ozone is a key trace gas that controls the temperature profile of the stratosphere and shields the surface from harmful UV radiation. Outside of the polar regions, total column ozone declined steadily throughout the last decades of the $20^{th}$ century, due to the emission of ozone depleting substances (ODSs) (WMO, 2014, 2018). ODSs are since declining as a result of the Montreal Protocol and its amendments; stratospheric ODS loading reached its maximum in the mid to late 1990s (Chipperfield



et al., 2017). The ozone decline stopped around the same time, but recovery of the ozone column is still not statistically
significant (WMO, 2018). Column ozone, however, is not the best metric to describe current stratospheric ozone changes, in
part because the ODS decline is not necessarily the primary driver of ozone trends. Increasing emissions of greenhouse gases
(GHGs) are expected to cool the stratosphere and accelerate the Brewer-Dobson circulation (BDC) (e.g., WMO, 2018), thereby
changing the background conditions and complicating the concept of ozone recovery. The impact of GHG-induced changes is
variable across the stratosphere. Stratospheric cooling slows temperature-dependent reaction rates, leading to reduced ozone
destruction in the upper stratosphere where the lifetime of ozone is short. In the lower stratosphere, accelerating tropical
upwelling and the balance of changes to the various branches of the BDC are the dominant controls on ozone concentrations.
The interplay of these changes with the ODS decline leads to disparate trends across the ozone column, as discussed below.

In the upper stratosphere, ozone is increasing at a statistically significant rate of 1-3 % decade$^{-1}$ (Ball et al., 2017; Sofieva
et al., 2017; Steinbrecht et al., 2017; Bourassa et al., 2018; WMO, 2018; Petropavlovskikh et al., 2019; Godin-Beekmann
et al., 2022). The confidence in the positive trends is greatest in the Northern Hemisphere (NH) mid-latitudes, while in the
tropics and in the Southern Hemisphere (SH) mid-latitudes trend magnitudes are generally smaller and nearer the 2-sigma
significance level (Petropavlovskikh et al., 2019). Both the ODS decline and the upper stratospheric cooling caused by increased
GHG emissions limit ozone destruction, and these two factors likely contribute about equally to the observed ozone increases
(WMO, 2014, 2018). In the lower stratosphere, the picture is more complicated. In the tropics, chemistry-climate models
(CCMs) predict ozone decline on the long term due to enhanced tropical upwelling (Dhomse et al., 2018; Morgenstern et al.,
2018). This agrees well with observations (Ball et al., 2020; Dietmüller et al., 2021), although there is large variability in both
models and observations, and not all satellite datasets indicate significant negative trends in the tropical lower stratosphere
(Petropavlovskikh et al., 2019). In the mid-latitudes, there's evidence that ozone might have continued to decline since 1998
(Ball et al., 2018, 2019; Wargan et al., 2018; Orbe et al., 2020), contrary to CCM predictions. These negative trends more than
offset upper stratospheric ozone recovery (Ball et al., 2018), and the mechanisms behind mid-latitude ozone changes are still
unclear.

Using three satellite composite datasets, Ball et al. (2018) found consistent negative trends below 24 km (32 hPa), with the
most significant trends in the tropical and NH lower stratosphere. Based on results from a chemistry transport model (CTM),
Chipperfield et al. (2018) argued that the negative trends in the mid-latitudes are the result of large natural variability, and
that a large positive ozone anomaly in 2017 removed the apparent negative trend. Subsequently, Ball et al. (2019) showed that
even with the large year-to-year variability, NH ozone continues to show significant negative trends. The authors pointed to
non-linear interactions between the quasi-biennial oscillation (QBO) and seasonal variability as the driver of large interannual
ozone variability in the lower stratosphere. This is in agreement with Chipperfield et al. (2017), who showed that post-1998
lower stratospheric ozone variability is largely driven by dynamics. The negative trends in the NH lower stratosphere were
confirmed by reanalysis results as well, with enhanced isentropic mixing (Wargan et al., 2018) and a poleward expansion of
NH upwelling (Orbe et al., 2020) identified as probable causes. Although Orbe et al. (2020) argued, based on one CCM, that
simulations can produce negative trends in the NH lower stratosphere, Dietmüller et al. (2021) showed that the observations are





a highly unlikely realization of CCM results based on the full ensemble of simulations performed for the Chemistry Climate Modeling Initiative.

Accurate description of stratospheric ozone changes is challenging, in part due to natural variability and sampling inconsistencies between various datasets. In addition, trends can be sensitive to the fit methods and time periods considered. Linear regression is commonly used to assess ozone trends, while dynamical linear modelling (DLM) is a more complex alternative (e.g., WMO, 2018). Linear trends typically include two components, connected or independent, to model ozone decline and subsequent increase. These methods are sensitive to the start and end dates, as well as the inflection point or period

(Petropavlovskikh et al., 2019; Dietmüller et al., 2021). DLM trends, on the other hand, vary smoothly over time, and are therefore able to represent non-linear ozone changes (Ball et al., 2017). In addition, a change in the analysis time period only affects the years close to the end points, while the rest of the DLM trend curve remains stable (Ball et al., 2019). A further complication in the mid-latitude lower stratosphere is the occurrence of second tropopauses and tropopause folds. The location of the tropopause needs to be accounted for to avoid potential negative biases by tropospheric contamination (e.g., Randel

et al., 2007; Sofieva et al., 2014). As a result of these challenges, the magnitude, significance, and, in the lower stratosphere, even the sign of stratospheric ozone trends is still in question.

Here we combine ozone data from three satellite instruments: the Stratospheric Aerosol and Gas Experiment II (SAGE II), the Optical Spectrograph and InfraRed Imaging System (OSIRIS), and, for the first time, SAGE III on the International Space Station (SAGE III/ISS). We present an updated OSIRIS data product (version 7.2), and use the composite dataset (hereafter the

SOS dataset) to assess stratospheric ozone trends. The dataset is filtered by tropopause altitude, and trends are determined using both multiple linear regression (MLR) and DLM methods. We use the full near-global dataset (1984–2021) to determine ozone change since 2000, and focus on the DLM results to examine recent ozone variability. To improve the trend results, we apply a reanalysis-based sampling correction to the OSIRIS and SAGE III/ISS datasets. In addition, we assess trend significance by explicitly accounting for differences between the SOS dataset and the Aura Microwave Limb Sounder (MLS). We show that

ozone increase since 2000 is robust in the upper stratosphere at mid-latitudes, but recovery largely stopped in the NH during the last decade. In the lower stratosphere, significant negative trends are present in the tropics, while mid-latitude changes are generally negative but not significant. Finally, the data shows that using tropopause-relative coordinates largely reverses the significance patterns and renders the ozone decrease in the tropics insignificant. The datasets, the sampling correction implemented for OSIRIS and SAGE III/ISS, and the trend fitting methods are described in Sect. 2. The upper and lower

stratospheric trend results from the merged dataset are discussed in Sect. 3, and the conclusions are given in Sect. 4.

## 2   Datasets and methods

### 2.1   OSIRIS v7.2 ozone

The OSIRIS ozone product is retrieved from limb scattered sunlight (Degenstein et al., 2009), and the previous data version (v5.10) corrects systematic errors in the limb pointing knowledge of the instrument to remove a long-term ozone drift (Bourassa

et al., 2018). The v7.2 OSIRIS ozone used in this paper is an improved version of the v5.10 product. The retrieval algorithm has



been updated and optimized, leading to minor changes in the ozone profiles. The changes to the ozone retrieval are described below, and the two data versions are compared in App. A.

Major changes to the retrieval algorithm include the use of a Levenberg-Marquardt algorithm (replacing the multiplicative algebraic reconstruction technique used for v5.10), and the implementation of a point spread function (PSF) correction. The
PSF correction reduces the effects of temperature-dependent blurring in the spectrograph optics, and it is performed for each individual scan by comparing the measured radiances at the top of the altitude range to a solar line atlas. The temperature of the OSIRIS optics varies seasonally, and shows a non-linear long-term decline as well (see Fig. 2 of Bourassa et al., 2018). The decline is largely due to a progressively lighter duty cycle and the decay of the orbit as the instrument ages. Another change to the retrieval is the use of temperature and pressure profiles from the Modern-Era Retrospective Analysis for Research and
Applications, Version 2 (MERRA-2: Gelaro et al., 2017; Wargan et al., 2017), instead of ERA-Interim (ERA-I). Furthermore, the retrieval now uses the 'DBM' ozone cross-section (Daumont et al., 1992; Brion et al., 1993; Malicet et al., 1995), the lower bound detection algorithm has been improved, and the iterative procedure was optimized to increase the number of profiles that converge.

In addition to the improvements to the ozone retrieval itself, the input parameters have been updated as well. The retrieval
uses aerosol and $NO_2$ number density profiles, both retrieved from the same OSIRIS scan as that of the ozone retrieval. A general feature of OSIRIS v7.x data are that the three products (ozone, $NO_2$, and aerosol) are now processed consistently and share the same version number. The aerosol retrieval, specifically, has been updated since the version used in the v5.10 ozone product. The new retrieval is based on a Levenberg-Marquardt algorithm, and the cloud flagging and lower bound selection have been updated as well (Rieger et al., 2019). The pointing correction implemented for ozone (Bourassa et al., 2018) is now
also included in the aerosol retrieval.

For the purposes of this study, only descending node (∼06:00 local time) OSIRIS measurements are used. There are biases between ascending and descending node ozone data, and the precession of the polar orbit changes the distribution of the measurements. This leads to loss of coverage at the ascending node, and systematic changes between ascending and descending node data over time (Adams et al., 2014). The large majority of the OSIRIS measurements are at the descending node.
Comparisons to MLS v4.2 ozone indicate that relative anomalies from the descending node data are mostly stable (within 2 % decade$^{-1}$; App. B), with the exception of regions with sampling biases. These sampling issues are addressed in the next section.

## 2.2   OSIRIS sampling correction

The analysis in this paper relies on monthly zonal mean (MZM) data. The latitude coverage of OSIRIS varies throughout the
year, and so MZM values at high latitudes are not always representative of the true center of the month and the latitude bin. This is a problem mainly at the edge of the latitude coverage, where measurements might only cover the first or last few days of the given month. Combined with the decreasing number of OSIRIS measurements in recent years, these sampling biases could lead to spurious trends in the MZM dataset (e.g., Damadeo et al., 2018) that need to be accounted for. To this end, we apply a sampling correction to the OSIRIS v7.2 data prior to the creation of the SOS composite.





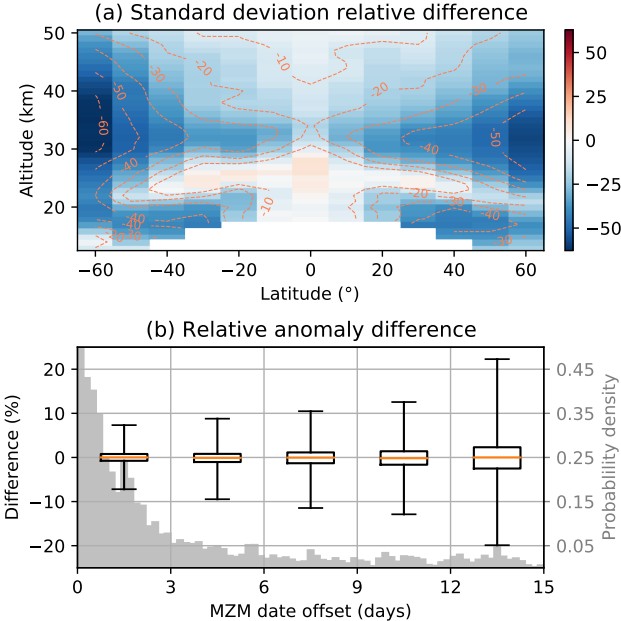

**Figure 1.** Effect of the sampling correction on OSIRIS v7.2 ozone. (a) Relative difference (in percent) of the mean ozone standard deviations. The standard deviations are calculated in monthly $10°$ latitude by $1°$ longitude bins, averaged across longitude and time, and then subtracted (corrected minus uncorrected). (b) Relative anomaly differences as a function of how far the mean date of the MZM is from the middle of the month. The box and whisker plots represent 3 day bins, with the whiskers extending from the $1^{st}$ to the $99^{th}$ percentile (outliers are not shown). The gray histogram shows the distribution of all MZM data along the same x-axis.

The sampling correction is performed using ozone profiles from MERRA-2. Ratios of ozone are calculated using the MERRA-2 profile interpolated to the time and geolocation of each OSIRIS measurement and a reference profile at the middle of the month and latitude band, but at the same longitude as the OSIRIS profile. Multiplying each the OSIRIS profile with the corresponding ratio transfers the ozone profiles to the middle of the month and latitude band, removing variability along the two axes that might not be fully sampled in the MZM. The correction does not attempt to remove longitudinal variability (the

dominant variability in the MZM), which is well sampled by OSIRIS. Final MZM values are calculated using the corrected OSIRIS profiles. The sampling correction relies on the accuracy of the MERRA-2 seasonal cycle for ozone, which is good in the stratosphere (Davis et al., 2017; Wargan et al., 2017). The method includes a simple diurnal correction as well, since the local time of the MERRA-2 reference profiles is fixed to noon, whereas average descending node local times for OSIRIS are between 6–8 am. It should be noted that the ozone diurnal cycle is not represented in MERRA-2 before 2004, i.e. before day

and night measurements from MLS were available for assimilation (Wargan et al., 2017).

    The effect of the sampling correction on the OSIRIS data is shown in Fig. 1. As the correction attempts to center the data along the latitude and time axes within a given month, Fig. 1a shows the change in the mean standard deviation along those dimensions. Standard deviations of the corrected data are reduced across nearly the entire stratosphere, indicating that





the sampling correction achieves the desired outcome. The largest changes (relative differences up to 40–60 %) are seen at
higher latitudes and in the lower stratosphere, where OSIRIS sampling is less dense. Figure 1b illustrates the change in relative
anomalies as a function of how far the calculated mean date of each monthly data point is from the middle of the month. Most
data points are within three days of the center, but a long tail extends across all possible offset values. The box and whisker
plots show that the sampling correction results in larger changes for months that are not sampled well, while the well-sampled
months are adjusted only minimally. The median change remains zero regardless of the date offset, indicating that there is no
overall sampling bias in the data. Changes in sampling over time still introduce spurious trends, however, and the sampling
correction successfully reduces temporal differences between the SOS data and MLS, as shown in App. B.

## 2.3 The SOS composite

For the SOS composite ozone time series, we use the sampling-corrected SAGE II v7.0 data (Damadeo et al., 2013, 2018) and
the SAGE III/ISS v5.2 data (Wang et al., 2020, for v5.1), alongside the sampling-corrected OSIRIS v7.2 ozone. The SAGE
II dataset extends from 1984 to 2005, while SAGE III/ISS and OSIRIS coverage begins in 2017 and in 2001, respectively,
and continues to the present (2021). All three instruments provide ozone number density profiles on an altitude grid, so unit
conversions are not necessary. All satellite profiles are interpolated to a common 1 km altitude grid. Since SAGE III/ISS
sampling is sparse, we apply the same sampling correction to the SAGE III/ISS ozone data as that described for OSIRIS in the
previous section. We use deseasonalized MZM relative anomalies to assess ozone trends. Prior to the anomaly calculation and
merging, we filter profiles from all instruments according to the tropopause altitude, as described below.

To ensure that only stratospheric measurements are included in the merged dataset, ozone profiles from all three instruments
are filtered by the tropopause altitude prior to merging. We calculate the lapse rate tropopause (WMO, 1957) for each ozone
profile using the corresponding MERRA-2 temperature profile. The first tropopause is defined as the altitude where the lapse
rate first decreases to 2 K km$^{-1}$, provided that the average lapse between this level and higher levels within 2 km is also below
2 K km$^{-1}$. For second tropopauses, the same threshold of 2 K km$^{-1}$ is used (instead of 3 K km$^{-1}$ in the original definition),
after Randel et al. (2007). If multiple tropopauses are present, the second tropopause is used as the limit, and data up to and
including the altitude level that contains the tropopause are discarded. The same tropopause filter is applied to the MERRA-2
reference ozone profiles used in the OSIRIS sampling correction (Sect. 2.2). Lapse rate tropopause altitudes from MERRA-2
agree well with those from radiosondes (Xian and Homeyer, 2019) and radio occultation data (Tegtmeier et al., 2020). Existing
biases (within 200 m) are small compared to the altitude resolution of the satellite datasets.

After applying the tropopause filter and the OSIRIS sampling correction, the datasets are merged as in Bourassa et al.
(2014, 2018). First, we calculate the MZM time series from the tropopause-filtered data, using a minimum of five measurements
per month, on a 10° latitude by 1 km altitude grid (grid centres of 60° S to 60° N, 13–50 km). SAGE II and SAGE III/ISS
data are then bias-corrected using OSIRIS as the baseline, and the deseasonalized anomalies are averaged. Deseasonalization
is performed independently for each instrument to account for any differences in the seasonal cycles. The relative anomalies,
used in the trend fitting, are calculated with respect to the entire time period, and so the time series might differ slightly if only
a subset of the data are considered.





For the purposes of this paper, the boundaries of the upper and lower stratosphere are considered to be 33 km and 23 km (grid centres, inclusive). Similarly, 'tropics' refers to the 20° S to 20° N latitude bins. Due to the tropopause filter, the number

of measurements in each MZM bin decreases sharply at low altitudes. To avoid fitting sparse time series, plotting limits are set to the larger of 12.5 km and the first altitude bin edge that is above the mean plus one sigma first tropopause altitude.

### 2.4  Trend analysis

We use both MLR and DLM to model long-term changes in stratospheric ozone. For both fit methods, known drivers of ozone variability need to be accounted for. To that end, we include the following regressors: the El Niño Southern Oscillation

(ENSO) from the multivariate ENSO index version 2 (without lag) (Wolter and Timlin, 2011), the first two principal component terms of the QBO (accounting for 92 % of the variance; https://www.geo.fu-berlin.de/met/ag/strat/produkte/qbo/qbo.dat), the F10.7 cm index representing the solar cycle (ftp://ftp.seismo.nrcan.gc.ca/spaceweather/solar_flux/daily_flux_values/fluxtable. txt), and latitude-dependent aerosol optical depth from the Global Space-based Stratospheric Aerosol Climatology (GloSSAC) (Kovilakam et al., 2020). The GloSSAC data (v2.1, 1979-2020) is extended to 2021 by extrapolating the last value. Since the

relative anomalies are deseasonalized, seasonal harmonics were not included for most regressors. We found, however, that including seasonal components for the QBO term reduced trend uncertainties with minimal impact on the trend values, and so the first two Fourier harmonics (representing annual and semiannual changes) were included for the two QBO regressors.

For the MLR fits, we use the LOTUS (Long-term Ozone Trends and Uncertainties in the Stratosphere) regression model (Petropavlovskikh et al., 2019). The (unweighted) regression is performed iteratively (Cochrane and Orcutt, 1949), and the

covariance matrix is updated for each iteration, taking into account the autocorrelation (Prais and Winsten, 1954) and data gaps (Savin and White, 1978). In addition to the regressors described above, the MLR model includes an independent linear trend component. To perform the MLR fit in a single step, five trend proxies are included: two slope and constant pairs to describe the linear change until January 1997 and after January 2000, and one constant to fit the transition period. Only the post-2000 trends are considered here, and significance is assessed at the 2-sigma level.

For the DLM fits, we use the *dlmmc* model (Alsing, 2019). A detailed description of the model and the methodology is given in Ball et al. (2017, 2019) and Laine et al. (2014). Briefly, the main difference between MLR and DLM is the inclusion of a smooth non-linear trend component in the DLM. The degree of trend non-linearity ($\sigma_{trend}$) is a free parameter in the model, and only the prior distribution of $\sigma_{trend}$ is specified (as a positive half-Gaussian with a standard deviation of $1 \times 10^{-4}$). In addition to the non-linear background trend, we include a first order autoregressive process (AR1) and the same set of regressors as for

the MLR. We keep the regressor coefficients constant in time, and so this component of the DLM is equivalent to the MLR without the linear trend. Since the relative anomalies are deseasonalized, overall seasonality is not included in the model. The posterior distribution of $\sigma_{trend}$ and the other DLM parameters is formed using Markov chain Monte Carlo sampling, with 10000 DLM samples (after an additional 2000 burn-in samples). The change in ozone is estimated as follows: for each DLM sample, the change is calculated between the yearly means for 2000 and 2021. The mean across the 10000 samples is taken as

the final result, while the probabilities of change are calculated from the integrals of Gaussian kernel-density estimates fitted





to the ensemble of values. The 95 % confidence level is used as the trend significance threshold. While the calculated DLM change is not necessarily equivalent to a trend in a linear sense, the two phrases are used interchangeably in this paper.

Both the MLR and the DLM fits are unweighted (use constant weights), since it is not evident what the proper uncertainties on the MZM values would be. The standard errors are dependent on ozone variability and sampling, and so the data are

heteroscedastic (and the true variances are not known). The LOTUS code (Petropavlovskikh et al., 2019) includes an optional heteroscedasticity correction using the fit residuals, based on the methods of Damadeo et al. (2014). The weighted MLR trend results, however, are sensitive to the exact correction method chosen, and the trends show unphysical oscillations with altitude. Weighted and unweighted DLM results only show minor differences, but for consistency we decided to use the unweighted results for both methods. We use the full SOS dataset (1984–2021) for all trend fits, although only the trends since 2000 are

considered. This ensures that the regressor fits are based on all available information. Additionally, using the middle of the time series as the reference, we avoid start date sensitivity for the DLM results (Ball et al., 2019). While both MLR and DLM results are discussed in Sect. 3, the main focus of the paper is on the DLM results.

Long-term stability is essential to accurately determine ozone trends. To assess the impact of instrument differences, we use MLS v4.2 ozone (Livesey et al., 2020) as the reference. We aim to answer the question: how does ozone trend significance

change if potential drifts are taken into account? To illustrate the change in significance, we take advantage of the distribution of non-linear trends provided by DLM. As described in App. B, we fit the SOS–MLS relative anomaly differences using DLM, and subtract the resulting mean trend from each individual SOS trend sample. We then use this adjusted SOS trend distribution to recalculate the confidence intervals for the 2000–2021 ozone change. Comparing the original and adjusted confidence intervals helps to identify regions where ozone trends are robust regardless of any instrument differences, and where the results need to

be interpreted with caution. Plots of the change in trend significance are presented alongside the original SOS trend results in Sect. 3.

## 3 SOS ozone trends

### 3.1 Upper and middle stratospheric ozone recovery

Ozone trends from 2000 to 2021 determined from the SOS dataset are shown in Fig. 2 for both MLR (Fig. 2a) and DLM

(Fig. 2b). It is immediately clear that the two methods yield similar results across the upper (>32 km) and middle (24–32 km) stratosphere, in terms of trend distribution and significance. Both methods show widespread significant positive trends, indicating robust ozone recovery since 2000. The values are largest at SH mid-latitudes, reaching over 7 and 6 % for MLR and DLM, respectively, at 50° S, while trends in the NH are slightly less positive (up to 6 and 5 %, respectively). Areas of near zero trends are present above 45 km (at both northern and southern mid-latitudes), at ±60°, and in the NH middle stratosphere.

Based on the DLM results, upper stratospheric ozone shows a significant increase of 2–6 % since 2000. Assuming linear changes, these values translate to trends of 1–3 % decade$^{-1}$, in agreement with previous studies (WMO, 2018). Changes in the middle stratosphere are smaller, at 1–3 % where positive trends are significant. MLR trend values are generally larger than the DLM results, by as much as 1–2 % in the tropics below 40 km and above 40 km at mid-latitudes. Differences elsewhere are



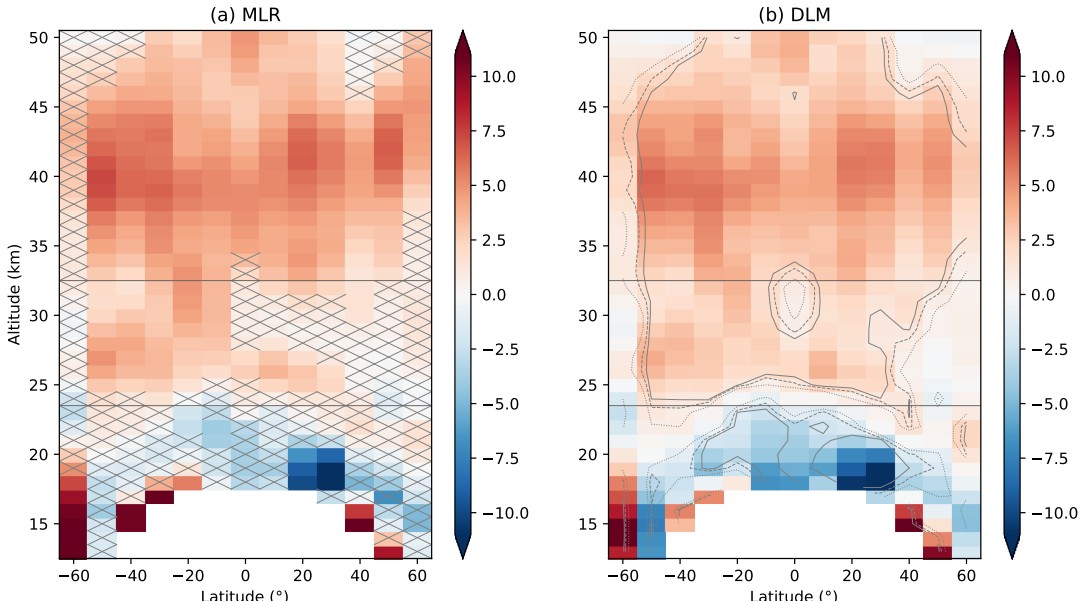

**Figure 2.** Percent ozone change from 2000 to 2021 using (a) MLR and (b) DLM. The cross-hatching in (a) represents lack of statistical significance. The MLR slopes and uncertainties were scaled to the 2000–2021 time period. The horizontal lines delineate the stratospheric regions, while the dotted, dashed, and solid contours in (b) show the 80, 90, and 95 % confidence intervals, respectively.

below 1 %. The large differences in trend values and the overall positive bias are likely due to the assumption of linearity in
MLR.

Linear ozone change is a good estimate for much of the upper stratosphere, as shown in Fig. 3. The figure shows the derivative of the non-linear DLM trend, i.e. the instantaneous rate of change for ozone, over the entire 1984–2021 period for each latitude bin. From 50° S to 30° N, ozone shows the expected structure of a linear decrease followed by a linear increase. The turnaround dates occur around 2000 or later, with significant variability across altitude. Turnaround dates are even more
variable in the middle stratosphere, although the rate of change is much smaller at those altitudes (especially in the pre-2000 period). The variable turnaround dates highlight one disadvantage of MLR, where the turnaround period is a fixed parameter that leads to endpoint anomalies in the trend results (e.g., Petropavlovskikh et al., 2019). DLM trends, on the other hand, are able to represent non-linear ozone changes, such as those at 40–60° N. At these latitudes, upper (and to a lesser extent middle) stratospheric ozone shows an initial recovery after around 2000, and a slight decrease since 2012–2015. Above ∼45 km, both
positive and negative changes are small, resulting in the area of no significant trends in Fig. 2b. At lower altitudes, the initial ozone recovery was more pronounced, and so trends since 2000 are still positive and significant. The decrease at 40–60° N is not significant when ozone change is calculated from the second turnaround point to the end of the time series, but the negative changes indicate a pause in ozone recovery nonetheless.



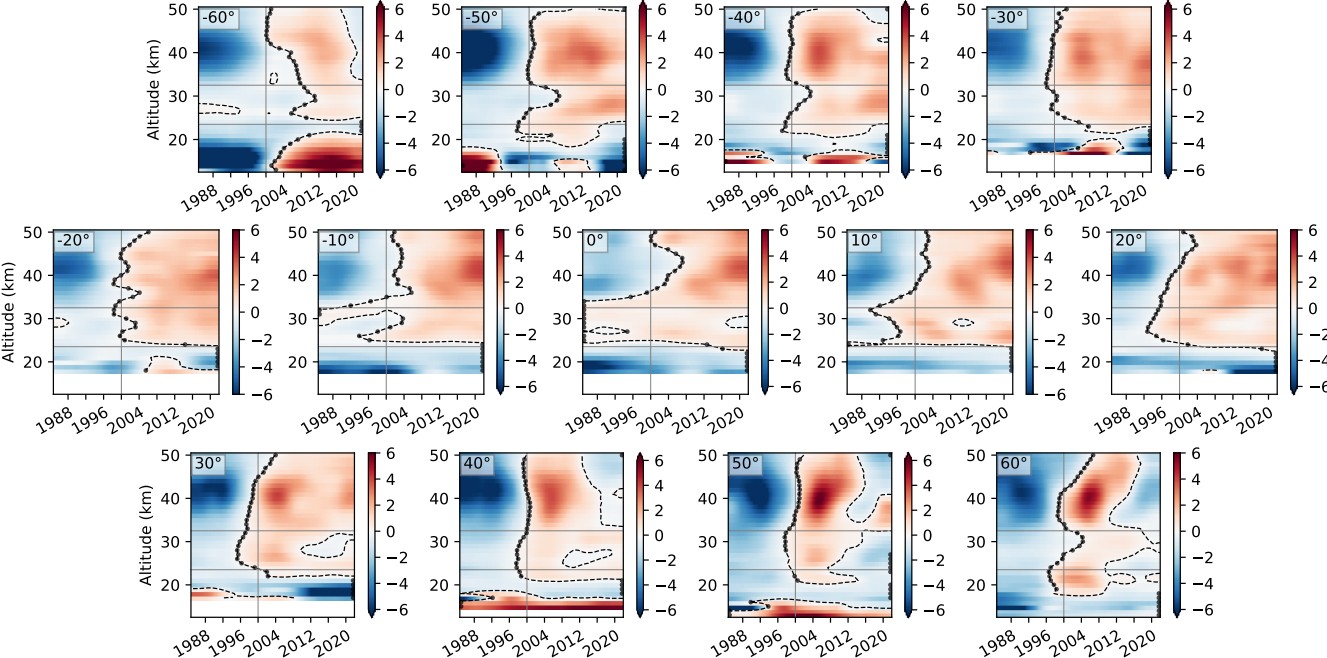

**Figure 3.** Slopes of the DLM trend results by latitude bin, scaled to units of percent per decade. The dashed lines show the zero slope contours, and the dots indicate the minima of the DLM trend fits. The vertical lines mark 1 January 2000, and the horizontal lines delineate the stratospheric regions.

As differences are present between the SOS data and MLS (App. B), we first examine the impact of these time-dependent
changes on the ozone trends. Figure 4 shows the change in significance when using the adjusted ozone trend distributions (Sect.
2.4). Most of the positive trends in the mid-latitudes remain significant at the 95 % level, and the region of significant ozone
increase in the NH expands slightly. The main change is the loss of significance for some of the positive trends in the tropics
and at high altitudes. These are the only regions in the middle and upper stratosphere where the combination of SOS minus
MLS differences (positive; Fig. B1c) and smaller trend values call trend significance into question. The upper stratosphere,
however, is also where the SOS–MLS differences depend strongly on the choice of reanalysis temperatures. Even so, some of
the trends in the tropics remain marginally significant (90–95 % confidence). Insignificant trends are still present near 50 km at
mid-latitudes, although these now fit into a pattern of insignificant trends globally at the same altitudes, similar to those shown
by Ball et al. (2019) near 1 hPa. While some negative changes (similar to Fig. 3) are still present in the NH in the adjusted
dataset, 2000–2021 trends are positive and significant in most of the NH, since ozone increase in the first half of the time period
masks recent non-linear changes.

To highlight more recent ozone trends, we calculate ozone change for the 2010–2021 period from the DLM fit results. We
chose 2010 as the reference since this is the last year when DLM ozone shows positive slopes for the entire upper stratosphere.
Figure 5 shows the 2010–2021 trend values, as well as the change in significance using the adjusted trend distributions. As


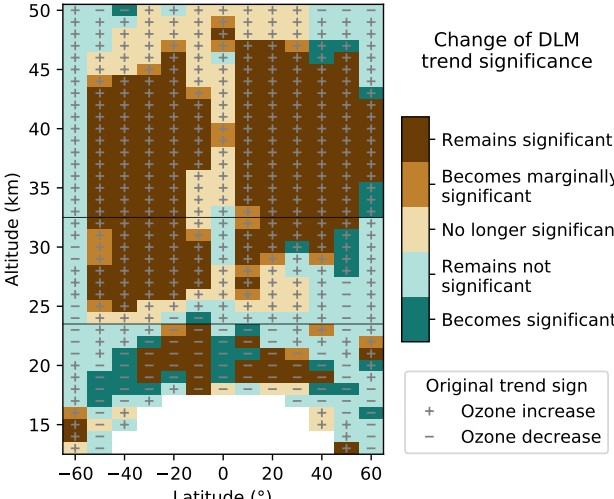

**Figure 4.** Change in significance of the 2000–2021 DLM ozone trends when using the adjusted trend distributions for the confidence interval calculation. Brown (green) colors indicate that the DLM trends are originally significant (not significant) with 95 % confidence. Different shades show if and how the significance changes. 'Marginally significant' indicates trends with 90–95 % confidence.

expected, significant positive trends are present across much of the stratosphere, but hemispheric asymmetries are more pro-

nounced than for the 2000–2021 changes. Above ∼43 km at mid-latitudes, trends are not significant in either hemisphere, although changes are more negative in the NH. At lower altitudes (including the middle stratosphere), SH ozone changes are significantly positive, with the exception of 60° S. At 40–60° N, however, there are no significant changes at any altitude, implying that ozone increase largely stopped in the NH mid-latitudes during the last decade. This conclusion holds even when differences with respect to MLS are considered. As shown in Fig. 5b, the adjusted trend distributions still result in signifi-

cant positive changes in the SH, and mostly no significant change in the NH. The differences are most striking in the middle stratosphere, where the asymmetry extends into the tropics as well.

Endpoint anomalies affect the exact distribution of significant trends, but the hemispheric asymmetry appears to be a general feature. To test the importance of the selected end year, we recalculated the SOS dataset and the corresponding DLM fit with data ending in 2017–2021. This changes the relative anomalies slightly, as the dataset available for the OSIRIS and SAGE

III/ISS seasonality calculation is truncated. In the middle stratosphere, the area of significant trends is reduced with decreasing end years, and so hemispheric asymmetries largely disappear for 2017–2019. In the upper stratosphere, however, a similar pattern is apparent for 2010 to 2017–2021 changes, although the contrast between hemispheres decreases for earlier end years. The reference year itself has little impact: considering ±2 years around 2010 results in similar significance patterns, and only changes trend magnitudes. The sensitivity of our results to the dataset end year is in agreement with Ball et al. (2019), who

performed similar sensitivity tests using DLM and found that the middle stratosphere shows the strongest end year anomalies.

Hemispheric asymmetries in ozone recovery have been reported elsewhere. The middle stratospheric differences are visible for the 2000-2021 period as well (Fig. 2), and have been observed in other datasets (e.g., Ball et al., 2019; Sofieva et al., 2021).





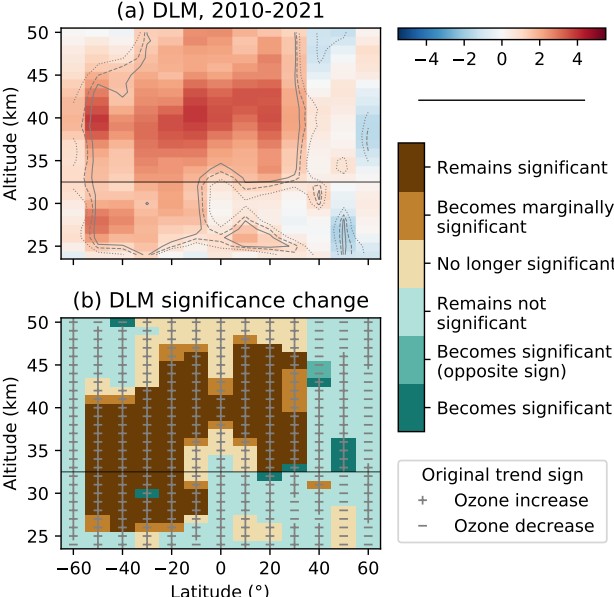

**Figure 5.** (a) As Fig. 2b, for 2010–2021 in the upper and middle stratosphere only. The color scale limits are halved given the shorter period. (b) As Fig. 4, for 2010–2021. An additional green shade indicates trends that become significant and change sign compared to the original (insignificant) trend.

In the upper stratosphere, where the longer period masks some of the asymmetries, Arosio et al. (2019) found broadly similar tendencies for 2012–2018. Highlighting underlying causes, Arosio et al. (2019) and Sofieva et al. (2021) both showed that
for 2003–2018, latitudinal trend structure in the NH is more variable compared to the SH, with larger positive trends over the North Atlantic and smaller or no trends over Siberia. Furthermore, Szeląg et al. (2020) showed that mid-latitude ozone trends exhibit significant seasonality as well, with the strongest mid-latitude ozone recovery during local winters.

Trend variability with seasons and latitude is likely related to the upper branch of the BDC, which transports ozone poleward from the tropics. There is evidence that the BDC, expected to accelerate due to increasing GHGs, is changing in a hemispher-
ically asymmetric way, which would then impact ozone trends. Mahieu et al. (2014) and Stiller et al. (2017) found large differences in hemispheric trace gas trends and attributed them to a relative slowdown of circulation in the NH compared to the SH, and a southward shift of the circulation pattern, respectively. More recently, Strahan et al. (2020) (and later Prignon et al., 2021) identified a low-frequency variability in the BDC with a period of 5–7 years that contributes significantly to inter-hemispheric anomalies of both trace gases and age of air. Both papers concluded, however, that air in the SH is getting younger
compared to the NH, indicating a relative slowdown of the BDC in the NH. How this might affect upper stratospheric ozone is unclear, especially since the exact magnitude of the ozone changes is difficult to establish. The recent pause of NH ozone recovery in the SOS dataset encompass only 1–1.5 cycles of the newly-identified BDC variability, and so longer time series will be necessary to properly account for the effects of natural cycles.



## 3.2 Lower stratosphere

The MLR and DLM fits show the largest differences in the lower stratosphere (<24 km; Fig. 2). In the tropics, both methods show consistent negative trends of 2-5 % and 2-6 %, respectively, for the 2000-2021 period. Trend magnitudes are similar above 20 km, while at 20 km and below MLR trends are less negative by as much as 1–4 %. Most of the trends are not significant for MLR, while the DLM results are significant (or nearly so) across most of the tropics. Figure 3 shows that from 20° S to 20° N, lower stratospheric ozone declined continuously since 1984 at almost every altitude, and current ozone levels are the lowest

in the entire DLM fit. As the negative DLM trends in the tropics are close to the significance threshold, even small changes between OSIRIS and MLS are sufficient to change significance. Fig. 4 shows that the adjusted trend distributions lead to a larger area of significant negative trends. This indicates that the ozone decrease in the tropics is robust, and likely stronger than SOS data show.

At mid-latitudes, MLR and DLM trends are generally not significant (Fig. 2). The largest differences in trend magnitude

occur in these regions: MLR trends are larger (more positive) in the SH, by up to 6 % at the lowest altitudes. In the NH, the differences are more variable and smaller (±2–3 %). MLR results show near-zero trends in the SH, with the exception of 60° S and a few altitudes near the tropopause, where significant positive trends (up to 12 % since 2000) are apparent. Positive trends in these regions are present in the DLM results as well, although at a smaller magnitude. Trends at 60° S are influenced by the polar vortex in austral spring, and so don't represent changes in mid-latitude ozone. Elsewhere in the SH, DLM trends are near

zero or negative, including a small region of significant negative trends at 30° S, and negative trends with 80–90 % significance at 40–50° S. In the NH, both methods show mostly negative trends below 20 km. These are only significant at 30° N, where large decreases are apparent. The magnitude (but not the significance) of these trends is explained by large negative anomalies in 2020–2021. In the DLM results, negative trends at the 80–90 % confidence level extend to 50° N. Positive trends are again present near the tropopause, and these are significant for both MLR and DLM.

Figure 3 shows that the history of ozone change in the mid-latitudes is highly variable, with the largest fluctuations in the SH. Change since 2000 is not necessarily a practical metric here, as the time series follow different patterns compared to higher altitudes. This, combined with the large variability, is responsible for the differences between the MLR and DLM results, as DLM is able to capture non-linear changes and multiple turnaround points. The end year of the time period affects results from both methods, although Ball et al. (2019) found that DLM trends in the lower stratosphere are not that sensitive to the end year.

This is in agreement with our DLM results. Significant negative trends are present in the tropics regardless of when the time series ends (2017–2021, see Sect. 3.1). At mid latitudes, the confidence in negative changes decreases with earlier end years. Trend variability is low, however: using twice the standard deviation of the 2017–2021 changes as the significance threshold, the region of significant trends approximately matches the 90 % significance contour in Fig. 2b. Differences compared to MLS have a larger impact on trend significance, and these differences are the largest at mid-latitudes. As in the tropics, the adjusted

trend distributions (Fig. 4) indicate that the area of significant negative trends is possibly larger than the SOS results show, especially in the SH. It is also clear that the few significant positive trends near the tropopause are not robust.





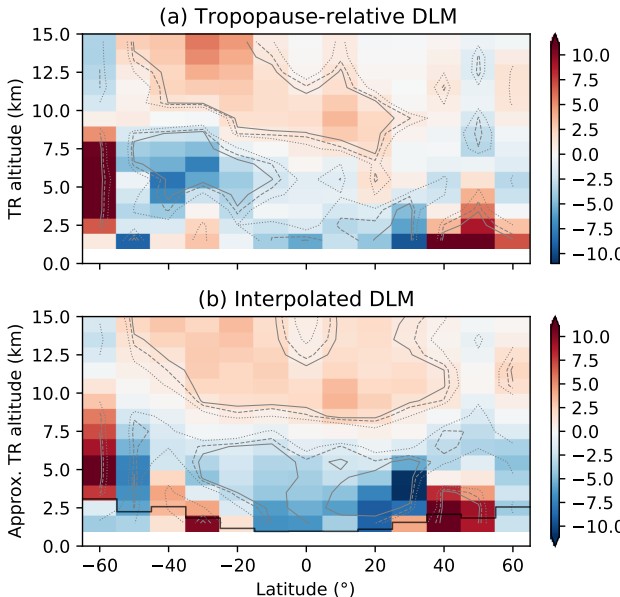

**Figure 6.** Percent ozone change from 2000 to 2021 using DLM. (a) Trends in TR coordinates. The bottom layer is empty by definition. (b) Results from Fig. 2b, interpolated to an approximate TR grid using the mean altitude of the first tropopause. The black line indicates the plotting limits in the original figure. Significance contours as in Fig. 2b.

The ozone decrease in the SOS dataset is similar to recent reports of lower stratospheric ozone decline in the literature. Using DLM and considering the 1998–2016/18 time periods, Ball et al. (2018, 2019) reported negative trends of similar magnitude in the tropics and in the NH, with smaller, less significant trends in the SH. The main composite dataset used by the authors (BASIC$_{SG}$) is largely based on MLS since 2005, and so App. B gives an indication of the differences we might expect. Using the BASIC$_{SG}$ dataset and linear trend estimates, Dietmüller et al. (2021) found very similar trend patterns, but much larger trend magnitudes, for 1998–2018. MLR trends from Petropavlovskikh et al. (2019) show similar results in the tropics and NH, with larger spread between datasets in the SH. While we use the same LOTUS model for MLR, time series in that study extend to 2016 only. Szeląg et al. (2020) examined seasonal trends in multiple composite datasets using a two-step MLR approach, and found significant negative trends in the tropics during spring and summer. In the NH mid latitudes, the authors found mostly negative trends, while SH trends were mostly positive. This is different from our results, although we include three additional years of data. It should also be noted that while Ball et al. (2018), Petropavlovskikh et al. (2019), and Szeląg et al. (2020) include a SAGE II – OSIRIS composite (with Ozone Mapping and Profiler Suite data), that dataset is substantially different from the SOS composite used here.

Ozone decline in the tropics is typically associated with the acceleration of the BDC. On the other hand, tropospheric warming also leads to increasing tropopause height and a general lifting of the stratospheric circulation (e.g., Vallis et al., 2015), which might explain some of the BDC (and ozone) trends in the tropics (e.g., Oberländer-Hayn et al., 2016). We rely on MERRA-2 temperatures to filter the SOS dataset (Sect. 2.3), and MERRA-2 tropopause height in the tropics is increasing at a





rate of 40–120 m decade$^{-1}$, for both dynamical (Wargan et al., 2018) and lapse rate (Xian and Homeyer, 2019) tropopauses.
Some significant trends are present at mid-latitudes as well; positive at northern and negative at southern latitudes. To examine
the impact of changing tropopause altitudes on the ozone trends, we repeat the DLM fits with the SOS dataset in tropopause-
relative (TR) coordinates. To do this, we interpolate each ozone profile to a TR grid (0.5–14.5 km; tropopause at bin edge) after
the tropopause filter step. Profiles are filtered up to the second tropopause (when present), but the first tropopause is set as the
TR grid reference for every profile. The rest of the processing (MZM calculation, data merging, DLM trend fits) is performed
as described in Sect. 2.

The results are shown in Fig. 6a, while Fig. 6b shows the original DLM trends, interpolated to approximate TR coordinates
using the mean height of the first tropopause in the SOS dataset. The results are similar above 8–10 km, indicating that TR
coordinates have little impact in the middle stratosphere and above. At lower altitudes, on the other hand, large changes are
apparent. In TR coordinates, most of the significant negative trends in the tropics disappear, and are replaced by smaller and
insignificant (but still negative) trends. One exception is the region 4–7 km above the tropopause at 10–20° S, where small but
significant ozone decrease is still present. At southern mid-latitudes, there is a large region of significant negative trends 3–4 km
above the tropopause, while in the NH, significant ozone decrease is present only at 30° N. Some positive trends are apparent
just above the mid-latitude tropopause, and these are large and significant in the NH. The variable trends near the tropopause
reflect the large fluctuations in lower stratospheric ozone anomalies. Variability as a function of end years is smaller, however,
especially in the tropics. TR trends remain mostly not significant in the tropics for dataset end years of 2017–2021, indicating
that this is a robust feature in the dataset. At mid-latitudes, the magnitude and significance of the observed ozone decreases
fluctuate with end year, while the positive changes remain mostly stable.

The TR ozone trends in Fig. 6a are similar to the results of Thompson et al. (2021), who used 1998–2019 ozonesonde data to
show that negative trends in the tropical lower stratosphere disappear in TR coordinates. The authors found that both seasonal
trend variability and differences between stations are greatly reduced in TR coordinates, and that no significant trends are
apparent in the ozone partial column 0–5 km above the tropical tropopause. Our results are also comparable to Wargan et al.
(2018), who used reanalysis data to show that at mid-latitudes, ozone is decreasing 0–10 km above the tropopause. The authors
found significant positive trends in the tropics (0–5 km in TR coordinates). This is different from our results, although Wargan
et al. (2018) included data up to 2016 only. Together these results indicate that ozone decrease in the tropics is in large part
controlled by increasing tropopause altitudes, as opposed to the acceleration of the BDC.

## 4   Conclusions

We used merged data from SAGE II, OSIRIS, and SAGE III/ISS to evaluate near-global ozone trends in the stratosphere. A
sampling correction based on MERRA-2 ozone was implemented for SAGE III/ISS and the new OSIRIS v7.2 ozone data,
and the method successfully reduces temporal differences between OSIRIS and MLS v4.2 ozone. The SOS dataset, filtered by
first and second tropopause altitudes, was fitted by both MLR and DLM methods to determine ozone change between 2000
and 2021. The two methods show similar results overall, both in terms of trend magnitude and significance. MLR changes,





however, are generally more positive than those from DLM, and this positive bias is related to the ability of DLM to model non-linear ozone changes. This is relevant for longer time series, as the relative importance of chemical and dynamical factors controlling ozone recovery changes over time and assumptions of linearity might break down.

Based on the DLM results, ozone increased significantly since 2000, by 2–6 % in the upper and 1–3 % in the middle stratosphere. These trends are significant across most of the upper stratosphere, and most of the southern and tropical middle stratosphere. The largest changes are present at mid-latitudes, and comparisons to MLS data indicate that these trends are robust with respect to deviations between the datasets. In the tropics and at high altitudes (near 50 km), trend values are smaller, and the choice of dataset might affect significance. The non-linear DLM fits reveal a recent pause of ozone recovery in the NH.

Since 2010, ozone at southern mid-latitudes increased significantly, while at northern mid-latitudes ozone has remained largely unchanged. The hemispheric asymmetry is independent of instrument effects, and is noticeable in the upper stratosphere using datasets ending in any of the past five years (2017-2021). Changes in ozone recovery might be related to interhemispheric differences and low frequency variability of the BDC, but longer datasets and better representation of dynamical variability in the trend fits will be required to fully separate the effects of natural cycles from long-term trends.

In the lower stratosphere, DLM results show that tropical ozone has decreased continuously throughout the entire 1984– 2021 period. This corresponds to an ozone loss of 2–6 % since 2000, significant across most of the tropics. At mid latitudes, significant ozone decrease is present only at $\pm 30°$. Ozone decrease with 80–90 % confidence, however, extends to $\pm 50°$, mainly in the 17–20 km altitude range. These results are largely independent of the dataset end year in the tropics, while trend significance varies more elsewhere, indicating that natural variability still hinders confident trend detection in the mid-latitude

lower stratosphere. Taking differences with respect to MLS into account shows that significant negative trends in the lower stratosphere are likely more widespread than the SOS dataset indicates, especially in the tropics and at southern mid-latitudes. Comparisons in the lower stratosphere, however, are less representative, as the sampling of both MLS and the SOS dataset is affected by the tropopause filter. Tropopause altitudes in the tropics are rising due to tropospheric warming. We found that in TR coordinates, most of the negative ozone trends in the tropics lose significance (but remain negative), and this feature is

independent of the dataset end year. These results show that tropical ozone changes are dynamically driven, and suggest that rising tropopause altitudes are more responsible for the ozone decrease than enhanced tropical upwelling.

*Code and data availability.* OSIRIS v7.2 ozone and the merged SOS dataset are available at https://research-groups.usask.ca/osiris/data-products. php. SAGE II v7.0 ozone (NASA/LARC/SD/ASDC, 2012), the SAGE III/ISS v5.2 data (NASA/LARC/SD/ASDC, 2017), and MLS v4.2 ozone (Schwartz et al., 2015) are available from the National Aeronautics and Space Administration (NASA). MERRA-2 data, used for

the OSIRIS and SAGE III/ISS sampling correction, are available at https://disc.sci.gsfc.nasa.gov/datasets?keywords=%22MERRA-2%22. SAGE II data were sampling-corrected according to Damadeo et al. (2018). The MLR code (LOTUS Regression v0.8.0, Petropavlovskikh et al., 2019) is available at https://arg.usask.ca/docs/LOTUS_regression/index.html. The DLM code (*dlmmc*, Alsing, 2019) is available at https://github.com/justinalsing/dlmmc (last access: 21 May 2021).



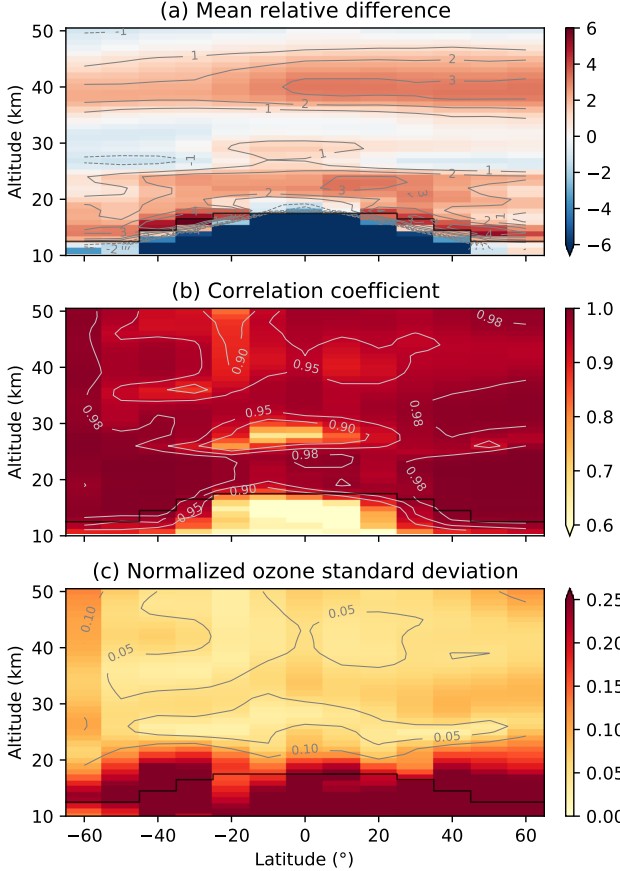

**Figure A1.** Comparison of OSIRIS v7.2 and v5.10 MZM ozone: (a) mean relative differences using v5.10 data as the reference, (b) correlation coefficients, and (c) standard deviation of the v7.2 MZM ozone time series, normalized by the overall mean in each altitude-latitude bin. The black line in each panel shows the selected tropopause cutoff; data below this line are not included in other figures.

## Appendix A: Comparison of OSIRIS v7.2 and v5.10 ozone

To show the changes in OSIRIS ozone between the v5.10 and v7.2 products, we compare the MZM values using only the descending node profiles that appear in both datasets. Since the v5.10 dataset is no longer processed, we only compare data up to the end of 2020. Figure A1 shows the overall statistics of the MZM comparisons. The two data versions are within 4 % in most of the stratosphere (Fig. A1a), with slightly larger differences (~5 %) near the tropopause. The v7.2 ozone values are generally larger than the v5.10 data. The positive offset in the upper stratosphere is most likely caused by the updated ozone cross-section, while in the lower stratosphere the offset is mainly the result of the updated aerosol retrieval (Sect. 2.1). The two data products show excellent correlation (Fig. A1b), with correlation coefficients ($R$) between 0.95 and 1 in most of the stratosphere. $R$ values in the mid-latitude lower stratosphere are consistently above 0.98. Lower $R$ values occur only in the





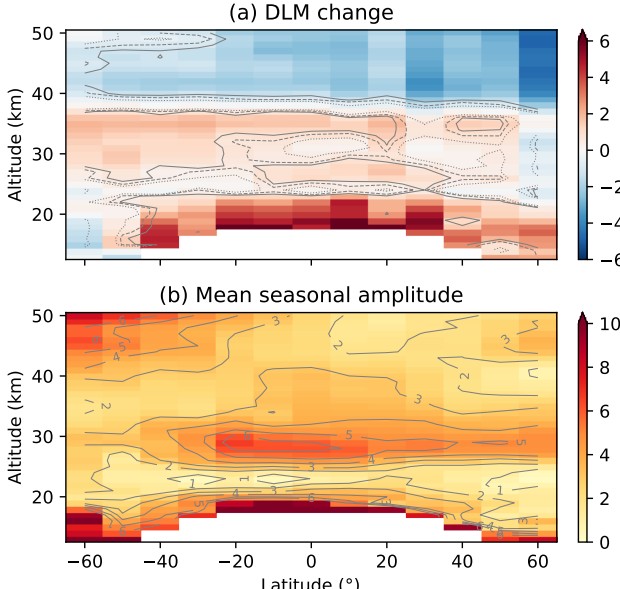

**Figure A2.** Relative differences in the OSIRIS v7.2 minus v5.10 MZM time series, fitted using DLM: (a) percent change between 2002 and 2020, and (b) amplitude of the fitted seasonal cycle. The dotted, dashed, and solid contours in (a) show the 80, 90, and 95 % confidence intervals, respectively.

tropical middle and upper stratosphere, where ozone variability is naturally low. These areas of reduced correlation correspond almost exactly to minima in the normalized standard deviation of the v7.2 ozone time series (Fig. A1c).

To examine any time-dependent differences between the two datasets, we employ the DLM methodology explained in Sect. 2.4 and App. B. We fit the relative differences of the MZM time series (as opposed to relative anomalies) to examine seasonal variability alongside long-term change. The change in the v7.2 minus v5.10 time series from 2002 to 2020 is shown in Fig. A2a. Small but significant drifts are apparent in most of the stratosphere, with patterns similar to those in the mean relative differences (Fig A1). Negative values above ∼38 km (up to 2–3 %) are due to a solar zenith angle dependent bias in the v5.10

data (Zawada et al., 2015), which results in ozone drifts given the small drifts in the descending node local time. The larger values in the lower stratosphere (up to 4 %, larger near the tropopause) reflect the larger differences between the two data versions at these altitudes. In the rest of the stratosphere, the small changes (below 2 %) are likely due to the combination of the PSF correction and the reduced dependence on solar zenith angles in the v7.2 data. Both of these factors lead to seasonal oscillations in the differences between the two datasets. As shown in Fig. A2b, the seasonality is largest around 25–35 km, and

above 40 km at mid-latitudes.

In summary, the v7.2 and v5.10 ozone products are in good agreement. The apparent differences are reasonable given the improved retrieval and updated retrieval inputs, and drifts between the datasets are small (equivalent to 1–2 % decade$^{-1}$). Comparisons of both data versions to MLS indicate that v7.2 ozone performs better in terms of stability.





## Appendix B: SOS temporal stability

In order to assess the stability of the SOS dataset, we compare the relative anomalies to those calculated from MLS v4.2 ozone data (Livesey et al., 2020). MLS provides measurements during both day and night with dense temporal and spatial sampling, and the ozone product is stable in the entire stratosphere (Hubert et al., 2016). The MLS ozone mixing ratios (on pressure levels) are converted to number density on the OSIRIS altitude grid using MERRA-2 temperatures and geopotential heights. Using reanalysis geopotential heights avoids ozone drifts caused by a drift in the MLS geopotential height profiles

(Hubert et al., 2016), although the choice of reanalysis temperature still affects the results, as discussed further on. The SOS minus MLS relative anomaly difference time series (2004/08 to 2021/12) is fitted using DLM, modified compared to the model used for ozone trends (Sect. 2.4). Here we include a seasonal term with annual and semiannual components, and remove the regressors entirely. The prior distribution of trend non-linearity is set to match that of the ozone trend fits, to better estimate the contribution of instrument differences to the ozone trends.

Two versions of the SOS data are included here: the first version uses the original, uncorrected OSIRIS and SAGE III/ISS data, and the second version includes the sampling correction for both satellite instruments. This second version is used in the main body of the paper, and the comparisons here show the effect of the sampling correction on the SOS dataset. Figure B1a shows the change in the SOS (uncorrected) minus MLS relative anomaly differences. The change is calculated from 2005 to 2021, the first and last complete year of measurements available in both datasets. Since the DLM is fit to the difference

of relative anomalies, the units represent the change in ozone that might be due to (non-linear) drift between the datasets. The lower stratosphere and 60° S stand out, while in the rest of the stratosphere there are small but significant differences. In general, the latter values are smaller than 3 % over the 2005–2021 period (below 2 % decade$^{-1}$), with the exception of the uppermost stratosphere at mid-latitudes. Positive changes are apparent in the tropics and in the SH upper stratosphere, while negative values are present in the NH. From the time series at 45 km (Fig. B1b), it is apparent that the negative values in the

NH are the result of changes since around 2015, while the positive drifts in the tropics and SH are linear across the entire time period.

As some of the drift between the datasets is likely due to the changing coverage of OSIRIS, the sampling correction (Sect. 2.2) should improve the results. This is shown in Fig. B1c, which is equivalent to Fig. B1a with the exception of the sampling correction for OSIRIS and SAGE III/ISS in the SOS data. Most noticeably, the large values in the mid-latitude upper strato-

sphere are greatly reduced, and improvements are seen in the lower stratosphere as well. The seasonal component of the DLM fit (not shown) indicates that the modeled seasonality in the relative anomaly differences is much smaller after the sampling correction is applied. Time series throughout the rest of the stratosphere show minor changes only, as these areas are well sampled in the SOS dataset. The overall fraction of significant trends decreases by 20 % in the entire stratosphere when the corrected dataset is considered. The negative trends in the NH are significantly reduced, and the time series (Fig. B1b) show

that the sampling correction reduces the recent negative differences compared to the uncorrected data.

OSIRIS contributes the majority of SOS data for the MLS comparisons, and the negative changes in the NH since 2015 coincide with an abrupt shift in the OSIRIS optics temperature trend. The hemispherically asymmetric negative changes might then



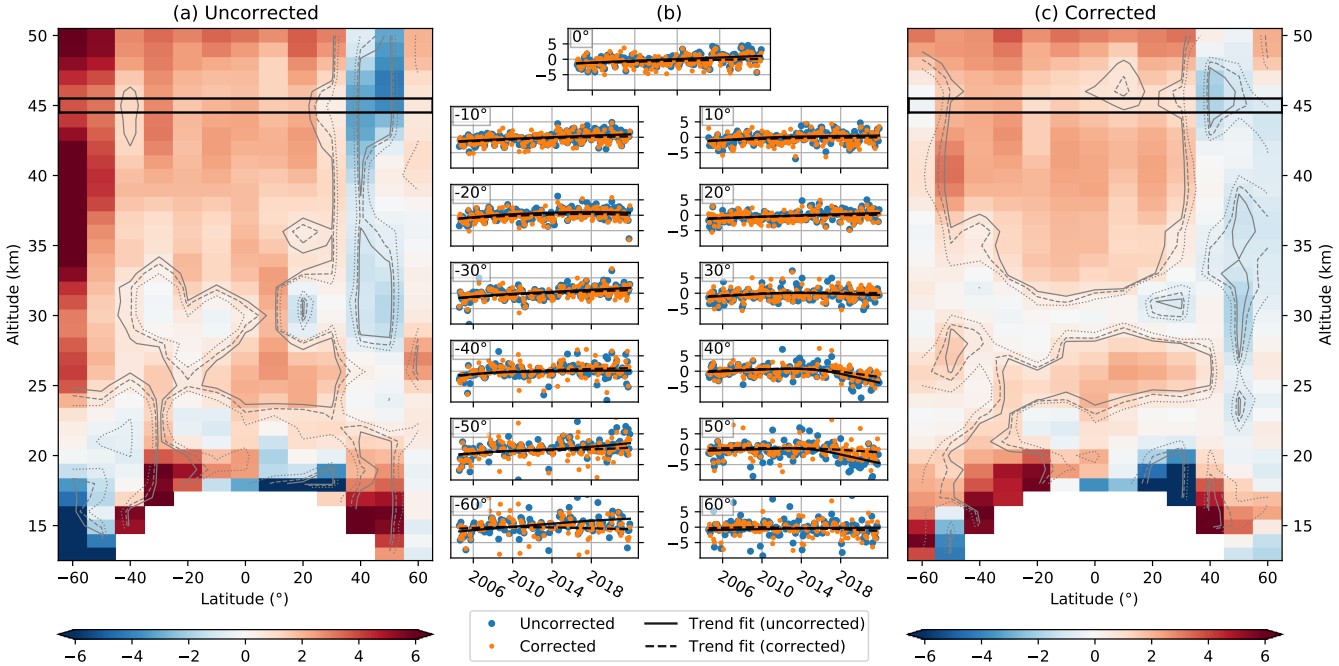

**Figure B1.** Differences of SOS and MLS relative anomalies. Panels (a) shows the SOS minus MLS relative anomaly differences for the SOS composite including the original OSIRIS and SAGE III/ISS data, while (c) shows the same for the SOS version that includes the sampling-corrected OSIRIS and SAGE III/ISS ozone. The units represent percent change in ozone between 2005 and 2021, while the dotted, dashed, and solid contours show the 80, 90, and 95 % confidence intervals, respectively. (b) Relative anomaly difference time series and DLM trend fits at 45 km.

be related to pointing issues caused by the changing thermal environment of the instrument. The pointing correction currently applied to OSIRIS data (Bourassa et al., 2018) mitigates these issues. The correction, however, uses a global daily average

shift, and latitude-dependent effects combined with changes in sampling patterns could lead to hemispherical differences in the MLS comparisons. SAGE III/ISS data (2017–present) masks part of these effects in the SOS composite, but the OSIRIS–MLS differences (not shown) are very similar to Fig. B1.

Remaining trends in Fig. B1b are largest in the lower stratosphere. There, however, the sampling of both SOS and MLS is affected by the tropopause filter (Sect. 2.3). This impacts the sampling correction (since the MERRA-2 reference profiles are

also filtered), and the comparisons themselves (since the assumption of perfect sampling for MLS breaks down). Furthermore, MLS v4.2 ozone shows small vertical oscillations in the tropical lower stratosphere (Hubert et al., 2016; Livesey et al., 2020), which might affect monthly data. Another factor to consider is the MLS data conversion, which is dependent on the choice of reanalysis temperatures. Recalculating the results in Fig. B1 using ERA-I (data up to 2019 only) leads to changes of 0.5–1 % in the upper stratosphere, with the largest differences (up to 1.2 %) in the mid-latitudes at 40–45 km. Drifts calculated using

the MERRA-2 conversion are more positive across the upper stratosphere, and so up to half of the positive changes discussed





above might be explained by the choice of reanalysis temperature. In the middle and lower stratosphere, results from the two reanalysis products agree well. This is not unexpected, as reanalyses generally show more differences in the upper stratosphere than at lower altitudes (e.g., Long et al., 2017). The impact of reanalysis temperatures, however, is largely uniform latitudinally, and thus doesn't explain any hemispheric asymmetries.

Overall, the sampling correction for OSIRIS and SAGE III/ISS successfully removes the largest drifts observed between SOS ozone and MLS v4.2 data. Minor trends remain across the upper stratosphere, with positive values equivalent to 1-2 % decade$^{-1}$ in the tropics and SH, and negative values of up to 1 % decade$^{-1}$ in the NH. Larger values are present in the lower stratosphere, where the sampling correction (and the comparison itself) is less effective. These results inform the SOS trend results presented in the main text, and are taken into account explicitly when assessing the significance of ozone trends.

As described in Sect. 2.4, SOS–MLS fit results are subtracted from individual SOS trend samples, and ozone change confidence intervals are recalculated using the adjusted trend distributions.

*Author contributions.*    KB performed the analysis and prepared the manuscript, with input from all co-authors. CR, DZ, and TW performed the OSIRIS retrievals and prepared the datasets. AB, DD, and ST supervised the project.

*Competing interests.*    The authors declare that they have no conflict of interest.

*Acknowledgements.*    The authors wish to thank the Swedish National Space Agency who enable the operation of the Odin satellite, the Canadian Space Agency for funding OSIRIS operation and data production, and the European Space Agency for including Odin as a Third Party Mission. SAGE ozone data were produced at the NASA Langley Research Center.



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
