# Peer review of "Stratospheric ozone trends for 1984–2021 in the SAGE II – OSIRIS – SAGE III/ISS composite dataset"

_Atmospheric Chemistry and Physics, 2022_

## Author Response (AR1)

**Response to Referee Comments**

The authors would like to thank the referees for their thorough reading of the manuscript, and for their helpful comments and suggestions. Our responses to each of the issues raised (in blue), along with the original referee comments (in black), are given below. Relevant changes made to the manuscript are described in each response. Line numbers always refer to the original manuscript.

**RC 1**

The paper describes a new merged ozone profile data set, based on SAGE II, SAGE III / ISS, and updated OSIRIS data. The resulting time series are analysed for ozone profile trends over the 2000 to 2021 period. The trends are largely consistent with previous findings and confirm the general understanding of ozone recovery and some climate change effects on the ozone layer. The findings also confirm the poorly understood continuing decline of ozone in parts of the extratropical lower stratosphere.

The paper presents new results, is based on solid data analysis and is well written. It fits well into the scope of Atmospheric Chemistry and Physics. I recommend publication with only very minor revisions.

My very minor textual suggestions are:

line 3: "remained ... constant ... due to changes" sounds weird. Suggest to replace "changes" by "the evolution"

The sentence now reads: "[…] mainly due to the evolution of lower stratospheric ozone"

line 9: "trends" might be a better word than "changes". Changes are certainly more complex than just trends.

Made the change.

line 13: Do you mean >80% confidence, or <80% confidence? Greater than 80% confidence would still be pretty good, and given the many sources of uncertainty might de facto not be much less confidence than greater than 95%.

We mean greater than 80% confidence, as shown in Fig. 2b. 95% is used as the significance threshold for the discussion, but the 80%, 90%, and 95% contours are shown in all the DLM figures.

line 17: Instead of "as well" I would say "even more". Dynamics play some role for ozone in the upper stratosphere, but in the lower stratosphere they are the major control.

Made the change.

line 18: the first "tropopause" needs to be "troposphere". The tropopause rises, but I don't think it warms.

Made the change.

line 23/24: There must be a better reference than Chipperfield et al. 2017 for the ODS turnaround.

Moved the reference to the WMO ozone assessments from the previous sentence (L22) to here and replaced the reference to Chipperfield et al.

On a different note, we also corrected the Chipperfield et al. reference on L53 (should refer to the 2018 paper instead of the 2017 one), and so the 2017 paper is no longer in the reference list.

line 26: replace "ozone" by "ozone total column", and then add something like "e.g. because tropospheric ozone columns are changing due to trends in precursor emissions (TOAR, 2020)".

Did not modify the sentence, as it is intended to lead from column ozone (used in the first few sentences) to stratospheric ozone. The rest of the paragraph discusses why ODSs are no longer the primary driver of ozone change in the stratosphere.

line 28: I think "process" would be a better word than "concept"

Changed to "[...] complicating the assessment of ozone recovery" to clarify the meaning of the sentence.

line 38: "reduce" instead of "limit". Also delete "likely" - we pretty much know this.

Made the changes.

line 39: "In the lower stratosphere ..." I would start a new paragraph here.

Made the change.

line 45: "but" instead of "and"?

Made the change.

line 58: Replace "a highly unlikely realization" by "at the extreme end of". The real atmosphere is never a realization of CCM results. And reality is also not unlikely. More likely, the CCMs are missing something, or we are at an extreme end.

Made the change.

line 60: add "at the one percent level" after "changes"?

Did not change the sentence, as it is intended to be a more general statement that leads into the discussion of trend models.

line 64: maybe add "chosen" before "inflection"?

Made the change.

line 66: I would delete the "therefore"

Made the change.

line 66/67: Suggest to drop the sentence "In addition .. 2019)."  The same kind of thing is true for linear trends. The middle stays the same, the end-points move.

The sentence intended to highlight that unlike piecewise or independent linear trends, DLM fits do not have a tunable parameter in the middle of the time series.

Replaced the sentence with:

"In the DLM, inflection points are fitted and not parametrized. Endpoint anomalies affect the ends of the time series only, while the rest of the DLM trend curve remains stable"

line 68: add "high variability, e.g. due to" before "the occurence". The tropopauses are just one aspect, a lot of complex stuff is going on.

Modified the sentence to start with "Another complication [...] is the occurrence of […]" to emphasize that tropopauses are just one piece of the puzzle.

line 71: "in question" or "unclear"?

Left as 'in question'.

line 75: add "using only data above the" before "tropopause"? I assume that is what is done, and would be a clearer description.

Replaced "by tropopause altitude" with "such that only data above the tropopause are considered".

line 79: "accounting for" or "considering"?

Left as "accounting for" to remain consistent with similar wording across the manuscript.

line 82: "largely reverses" or "changes"?

Left as 'largely reverses', since this refers to the shift of the significant negative trends from the tropics to mid-latitudes.

line 89: replace "to remove" by "which removes". I hope you are doing it right, and then the drift goes away.

Flipped the parts of the sentence to say that "[v5.10] removes a long-term ozone drift by correcting systematic errors in the limb pointing knowledge of the instrument".

line 126: add "MERRA-2" before "reference"?

Made the change.

line 127: "each the" -> "each"

Made the change.

line 129: "time and latitude" instead of "two"

Rephrased as "[…] along the two axes (time and latitude) [...]"

line 130: delete "in the MZM". You are not talking about variability of the MZMs, you are building the MZMs.

Replaced "in the MZM" with "in each zonal bin" to clarify the meaning of the sentence.

line 146: You mention it later in the next paragraphs, but it might be good to already say it here: "The same sampling correction is applied to the SAGE III/ISS data, and the SAGE II data are also sampling corrected".

Since this section focuses on OSIRIS data, we did not add references to the SAGE instruments here. The sampling correction for each instrument is discussed in the paragraph immediately following this section.

line 178: "estimate" instead of "model"?

Lefts as "model", since we feel that modeling is the appropriate description.

line 186/187: "coefficients" instead of "term" and instead of "regressors"

Reworded sentence to refer to "[...] including seasonal QBO components [...]". Left "regressors" as is since that terminology is used throughout the paper.

Figure 2 caption: State the confidence level used for hatching in 2a. I assume it is 95%. I would also prefer to have hatching in Fig. 2b - that would make the comparison between the two panels much easier.

Edited the caption to say that the hatching "represents lack of statistical significance at the 2-sigma level". While hatching in Fig. 2b would make comparisons simpler, it would also mask useful information. The DLM provides a more detailed uncertainty estimate, and we prefer to show the confidence intervals here (and on all other DLM plots).

line 348: add "-OMPS" after "OSIRIS"?

The sentence now reads:

"[...]include a SAGE II - OSIRIS - OMPS-LP (Ozone Mapping and Profiler Suite Limb Profiler) composite, [...]"

line 355: trends of what? Please clarify.

Added "in tropopause heights".

line 387: Is the positive bias related to the ability? Or is it related to a downturn / lower values modelled by the DLM (compared to linear trend)? I think this should be phrased better.

Left the sentence as is, since this part compares the two methods, and the underlying reasons (the lower modeled values) are discussed in the next paragraph.

line 389: I would replace "linearity" by "linear changes over time". Lots of things are still pretty linear.

Left as is, since the meaning is clarified by the previous sentence, and the phrase 'over time' is already used in this sentence.

line 404: "still hinders confident trend detection" It may do that forever. I'd rather say something like "masks possible trends"

Made the change.

line 409: "magnitude and significance" instead of just "significance"

Changed to "significance and magnitude".

line 411: replace "are more ... tropical upwelling" by something like "play an important role". I don't think you have shown anything about upwelling, and I am not sure you can really separate upwelling, tropopause changes, tropospheric changes, wave-driving, ... in your data.

Reworded the sentence to say that rising tropopause altitudes "are in part responsible for the observed ozone decrease in the tropics".

Fig. A1: Is the correlation for data with annual cycle, or for anomalies with annual cycle subtracted. Please explain / add.

All comparisons are performed using the MZM data (as stated in the caption of Fig. A1), and so the seasonal cycle is included. Amended the first sentence of App. A to emphasize that "[...] we compare the MZM values (not anomalies) […]".

line 444: I would rename this to "SOS comparison with MLS" Among other things, we don't really know how stable MLS is.

Made the change. See also the response to RC2; the intent is not to set MLS as a perfect reference, but to provide and extra piece of information by comparing the two datasets.

line 453: add "QBO, ENSO and F10.7" before "regressors"

Added the list of regressors in parentheses.

line 498: can you explain better what you mean by "inform"? I guess you take random samples of the SOS DLM trends, and samples of the SOS-MLS DLM trends, and then get a resulting "corrected" SOS trend distribution?

As described in Sect. 2.4, we take the mean trend from the SOS-MLS DLM fit, and subtract it from each individual SOS trend sample. This results in the adjusted trend distribution that we use to update the confidence interval calculations. We did not add more explanation in the appendix, as L500 already refers to the relevant section in the main body of the paper.

**RC 2**

**Summary:**

The authors describe, and to an extent assess, a new version of the OSIRIS ozone data set (v7.2). They then combine this data with ozone data from SAGE II and SAGE III/ISS to create a composite data set to be used for ozone trending analysis. This composite incorporates a sampling correction to attempt to mitigate the potential impacts sampling biases can have on trend analyses. Trend analyses are performed using, and comparing, both MLR and DLM methods with most of the analysis focused on the DLM results. The overall trend patterns are largely consistent with those from other studies over similar time periods. However, the DLM results show a curious second turnaround in the middle-to-upper stratosphere at northern mid-latitudes starting in the mid-to-late 2010s. The authors also demonstrate, through trend analysis in tropopause-relative coordinates, that dynamical effects, particularly a positive trend in tropopause height, are primarily responsible for continuous negative tropical ozone trends. Lastly, the authors consider the potential impact of instrument drifts on resulting trend significances by comparing the data composite to separate measurements from MLS.

Overall this paper is well-written, the work is well-thought out using commonly accepted techniques, and the conclusions are mostly reasonable. However, I do have some general questions and minor comments with respect to the methodology as well as to the overall structure of the paper that I would like to see addressed prior to publication.

**General Comments:**

The description of changes to the OSIRIS algorithm / data versions feels very disjointed between the text and appendix. It appears that this is the first time the v7.2 data are being used and so I believe the authors intend for this paper to also act as a data reference of sorts. However, the description of changes feels rushed and incomplete. If there's enough material, a separate paper detailing the changes and comparisons for validation purposes would make more sense, though some of the biggest influences seem to be already detailed in other papers so perhaps that is not the case. At the very least (i.e., in lieu of a separate paper), though this is a request outside of this work, I would recommend the authors also put together some sort of release notes with a more singular story regarding improvements (perhaps with some standard comparison validation figures) that could also be referenced for future works. In its current form, this paper does not tell a cohesive story regarding how OSIRIS data has changed between versions and what the impacts to the data quality are.

Version 7.2 of the OSIRIS ozone data is a very recent release, and this paper is indeed intended to serve as a data reference for v7.2. We've decided against a separate publication mainly since v7.2 is not a new product but an improved version of previous products. There are not enough changes/differences to justify a separate paper, and the manuscript already lists all the relevant updates. In addition, version numbers are incremented together for the ozone, $NO_2$, and aerosol products, which might result in a new version with no changes to the ozone algorithm.

Release notes and a version-by-version description of the changes (along with other useful information) are now available at https://arg.usask.ca/docs/osiris_v7/index.html. A link to this website has been added to the manuscript. The 'Changelog' tab on the website describes the differences between various data versions ($5.10 \to 7.0 \to 7.1 \to 7.2$), and specifies which product (ozone, $NO_2$, or aerosol) was updated.

We restructured section 2.1 in the manuscript to tell a more coherent story. We kept the format as a summary of changes (in order to avoid shifting the focus from the main purpose of the paper), but the content has been reorganized:

- The first paragraph now explains why there's a large gap between version numbers, and includes references to the release notes and to the appendix.
    - The sentences describing the shared version number (L104-107) are now in the first paragraph.
- The second and third paragraphs now focus on major and minor changes, respectively
    - The second paragraph was shortened to discuss only the Levenberg-Marquardt algorithm switch and the PSF correction.
    - The remainders of the second and third paragraphs were merged to discuss the other changes.

I think it's worthwhile to change how the QBO is incorporated into this work. The relatively recent series of ozone papers (specifically Ball -> Chipperfield -> Ball) demonstrated that the DLM's adaptable shape is more susceptible to influence from endpoint anomalies caused by potentially improperly capturing natural variability. I am glad to see the inclusion of seasonal cross-terms with the QBO, despite this being outside the original LOTUS framework, as the seasonal cycle modulates the frequency of the QBO, particularly at mid-latitudes. What I'd really like to see though is a change to the number of principal components / EOFs that are used. I know in the past I have recommended using, and this group in particular has used, more than the leading 2 EOFs. Prior to about 2015, this seemed like it may have only yielded a minimal change to the results. However, since that time there have been two major disruptions to the QBO and, perhaps more importantly, the leading 2 EOFs are no longer sufficient to capture all of the variability associated with the QBO. Anstey et al. (2021; DOI: 10.1029/2021GL093058) demonstrates that the leading 4 EOFs are necessary to capture these new features. I wonder what kind of an impact not adequately capturing the QBO has on the DLM's trend results and I think that it's time to start taking this effect into account as a standard practice. I am particularly curious to see if there is any change to the somewhat peculiar behavior of the DLM shown in Fig. 3 at mid-latitudes in the middle to upper stratosphere where the trends appear to turn around again.

We have run tests using four QBO principal components, but ultimately decided not to include those results in the manuscript. Briefly, the change in the fit results is minimal at the cost of massive increases in computation time for the DLM, dramatically reducing reproducibility.

Using either two or four QBO components, the DLM fits are essentially unchanged. The exact values for the 2000-2021 (i.e., Fig. 2) change vary slightly, but the overall behavior and the trend significance contours are very similar. The DLM slopes (i.e., Fig. 3) show the same results: the rates of change and the timing of the turnaround points is very similar. This holds in the northern mid-latitudes as well, and both versions of the DLM fit show the same structure and similar magnitudes for the second turnaround. To illustrate the similarities of the results, we include the original Fig. 5 (with the updated color scheme; see response to last comment) and the same plot using the DLM fit with four QBO components. This figure shows the largest changes when four QBO components are used.

[Figure]

*Illustration 1: Left: Fig. 5 of the original manuscript (two QBO components), with the updated color scale. Right: DLM fit with four QBO components.*

As seen in Illustration 1, the additional QBO components have a very limited impact on the behavior of the DLM fit. Both version show the same pause in ozone recovery in the northern mid-latitudes. The fits do change slightly, and this results in slightly more significant trends in the adjusted trend distributions at 30-40°N around 40 km. We'd argue, however, that both sets of results tell essentially the same story, and the original set of DLM fits is far easier to reproduce.

The DLM fits with four QBO components take over two weeks to run using 7 CPUs in parallel. This is nearly five times the time it takes to perform the original (already computationally intensive) fits. Given that we include seasonal harmonics for the QBO regressors, the current methodology is simply not suitable for more QBO components The seasonal components are arguably necessary since they do improve fit quality. For tests with no seasonality, increasing the number of QBO components from two to four results in negligible changes only. In addition, using two QBO principal components with seasonal terms is in line with the latest version of the LOTUS model (Godin-Beekmann et al, 2022; DOI: 10.5194/acp-2022-137).

We have added a sentence at L187 (Sect. 2.4) to say that "Including additional principal components of the QBO results in minor changes only, and so only the first two principal components are used here."

While stated in the paper, it is difficult to use the word "trend" when discussing the DLM results. The authors compute the difference between start and end years (am I interpreting that correctly?) and call that the trend (obviously with the proper unit conversion), but this means any meaningful decreases in the last year (or last few years) could yield significantly different results than the MLR that yields a rough average of the change over the time period. This means that the MLR is generally better than the DLM, at least for this calculation, at stating the overall trend results while the DLM is generally better than the MLR at stating the overall difference between any two years. I just think it's important to mention the caveats about the interpretation of results between the two methods if results from the two methods are going to be directly compared. Right now there is just the statement on Line 239 about there being a positive bias, which implies that the MLR is worse at representing the "trend" because it assumes linearity.

We chose the comparison of the DLM changes to the scaled MLR trends in order to balance some of these issues. The DLM 'trend' is indeed just the change between two yearly means, but it is compared to the MLR estimate of ozone change in the same interval. While exact comparisons are not possible, these metrics are useful to 1) show the approximate agreement of the methods, and 2) represent the DLM trend line as a single value. The evolution of DLM ozone is shown (Fig. 3) to aid in the interpretation of Fig. 2. The non-linear DLM trend is sufficiently smooth that only changes on the decadal scale have a large impact on the trend values, and using yearly averages instead of just the end point also mitigates the impact of endpoint anomalies.

To highlight the challenges of comparing MLR and DLM results, and explain why the DLM results are referred to as 'trends', we've expanded the relevant sentence (L206-207) into a short paragraph:

> "To compare ozone trends from the two methods, MLR trends are scaled to the time period of the DLM ozone change (2000-2021). The DLM values are referred to as 'trends' throughout the paper for simplicity's sake, although they represent the difference between two yearly means along the non-linear trend line. These values can diverge from the scaled MLR trends if the underlying DLM fit is sufficiently non-linear. To aid in the interpretation of the DLM results, both the trend values and the ozone rate of change over time are shown."

In addition, we rephrased the reference to a 'bias' at Line 239: "The large differences in trend values and the overall positive offset are likely due to the fundamental differences between the two methods."

The "adjusted" trend distributions using MLS data are a tad odd to me. I understand the desire to assess potential drifts in the data set, but, even with the improvements between data versions that specifically mention removing some potential drifts, the verbiage in the paper reads like the implicit underlying assumption with this is that OSIRIS data has a drift and MLS doesn't and that biases the trend results. Such a statement would make the reader question the utility of the SOS data set for trend analyses and wonder why one would use OSIRIS data instead of MLS data for a data composite, particularly data after 2015 (i.e., optics temperature dependency). It would be interesting to see if any drift is also visible

in just the differences between OSIRIS and SAGE anomalies over the roughly 4.5 years of data overlap as an independent verification.

The main goal of the SOS-MLS comparisons is to evaluate the sampling correction and to provide an additional metric to assess ozone trend significance. This is the reason why we compare the composite dataset (and not the individual instruments), and why we use the same MZM time series that might be used for trend studies (instead of pairing the datasets at the profile level). The comparisons shed light on the differences between the datasets, but as pointed out in Appendix B and in Sect. 3.1, direct comparisons (or the use of MLS data for this composite) are challenging given the uncertain conversion of MLS VMRs to number density.

To fine-tune the discussion and place the two datasets on more equal footing, we made the following changes:

- L219-220: rephrased as comparison instead of using MLS as the reference; now refer to "potential drifts in the datasets"; and added a sentence to say that "With the assumption that the truth lies somewhere between the SOS and MLS time series, we can construct an additional significance metric for the ozone trends.".

- L444: the title of Appendix B is now "SOS comparisons with MLS" (see also response to RC1).

- L445: the goal of the comparisons is now stated as "In order to evaluate the sampling correction and help assess ozone trend significance, we compare […]".

- L467: the opening sentence now says "much" instead of "some" of the drift is related to sampling issues.

- L476: now refer to the "remaining" negative differences, as most of the negative trends are removed by the sampling correction.

- L482: added an extra sentence to explain that "[...] the sampling correction likely doesn't remove all sampling issues, and so some of these negative differences might still be sampling-related.".

Unfortunately the SAGE III/ISS time series is too short for meaningful comparisons to OSIRIS. With the current methodology, the fitted trends are not significant, and the trend patterns mostly resemble random noise. The SAGE III/ISS data are helpful for filling in the gaps in the OSIRIS sampling pattern, but not yet sufficient for trend studies on their own.

Operating under the assumption that the drift between OSIRIS and MLS data is entirely a result of anomalies in the OSIRIS instrument brings up some additional questions. The difference between Fig. 2b and Fig. 5a obviously stems from the fact that the SOS data shows a second turnaround that begins in the early-to-middle 2010s, particularly at northern mid-latitudes. Comparing this (mental) difference to Fig. B1c seems to suggest that this second turnaround feature present in the SOS composite may be, at least partly if not entirely, the result of instrument drift and not anything physical. Figure 5b shows that accounting for these MLS differences doesn't change the trend significance, but I would also be

interested to see another category of shading on this plot, namely "Remains not significant (oppositely sign)", and how it might manifest at NH mid-latitudes.

The SOS-MLS differences contribute to the changes observed in the norther mid-latitudes. The main reason we included Fig. 5b is to show that accounting for the instrument differences doesn't remove this feature, indicating that the pause of ozone increase is present in the underlying time series. We have run tests where the OSIRIS data is replaced by MLS in the composite, and the resulting DLM slopes show patterns similar to Fig. 3. The test dataset shows a clear slowdown of ozone recovery (near zero slopes), and negative slopes are present at 50-60°N as well (covering smaller areas compared to Fig. 3). We don't discuss these tests in the manuscript as using MLS data for ozone trend fits is outside the scope of this project.

As we only claim that there is a pause in ozone recovery in the NH, we believe Figs. 4 and 5b are sufficient to show that this pause is present regardless of any instrument differences. Regarding the additional category in Fig. 5b, we include an updated figure here to demonstrate the changes, but we decided to keep the original categories for the manuscript (for reasons explained below).

[Figure]

*Illustration 2: Identical to Figure 5 in the manuscript, with the exception of the color scale and one additional significance category ("Remains not significant (opp. sign)").*

It should be noted that while the 2021 slopes (Fig. 3) are almost all negative in the NH, the 2010-2021 changes (Fig. 5) are not necessarily negative. Where the changes are negative and not significant, half the bins change sign with the adjusted dataset (Illustration 2). The opposite happens as well, however; at 48-50 km at 60°N, the sign changes from positive to negative.

Adding the new category to fig. 5b could be misleading, however, since the point is that the changes are not significant (i.e., consistent with zero). We included the original signs in order to make comparisons with the trend figures easier, but the signs are only really meaningful for significant trends. Especially in the NH, the changes are small, and so the difference between positive and negative

changes might be negligible even if the changes themselves were significant. Since significance contours wouldn't quite fit on this figure, we prefer to have only the two categories for not significant data ("remains not significant" and "no longer significant").

**Specific Comments:**

L023: "stratospheric ODS loading reached its maximum in the mid to late 1990s (Chipperfield et al., 2017). The ozone decline stopped around the same time …"

At least in the upper stratosphere

Left the sentence unchanged, since both the sentence on L21 and the second half of the current sentence refer to total column ozone.

L024: "but recovery of the ozone column is still not statistically significant (WMO, 2018)"

This is still true almost everywhere except we're starting to see significant positive trends at southern mid-latitudes (see Fig. 4 of Weber et al., 2022; DOI: 10.5194/acp-2021-1058).

Added a citation to Weber et al at the end of the sentence: "[…], with the exception of emerging significant positive trends at southern mid-latitudes (Weber et al., 2022)". We initially didn't include the paper as it was not yet accepted by the time this manuscript was submitted.

L029: "stratosphere. Stratospheric cooling slows temperature-dependent reaction rates, leading to reduced ozone destruction in the upper stratosphere where the lifetime of ozone is short. In the lower stratosphere, accelerating tropical upwelling and the balance of changes to the various branches of the BDC are the dominant controls on ozone concentrations."

I would add a reference for each sentence (at least for the second one).

Added a reference to Chipperfield et al. (2018) who show that lower stratospheric ozone is largely under dynamic control.

L044: "These negative trends more than offset upper stratospheric ozone recovery"

You mean in terms of their impact on the total stratospheric column and the ozone layer as a whole.

Yes, and we think the context is clear. We did modify the sentence to say that "negative trends would more than offset upper stratospheric ozone recovery". See also response to RC1.

L125: "The sampling correction is performed using ozone profiles from MERRA-2."

Is there any concern of introducing a bias if the spatial (i.e., meridional) gradient in ozone is not adequately captured by the model? Also, it's interesting how the standard deviations appear to increase in the tropical lower stratosphere in Fig. 1. Why do you suppose it gets worse?

Small biases are possibly present, although this is likely not a significant issue since MERRA-2 ozone compares well to observations in the stratosphere (Davis et al., 2017; DOI:10.5194/acp-17-12743-2017 and Wargan et al., 2017; DOI:10.1175/JCLI-D-16-0699.1, as noted at L132). In addition, these issues would only affect the trend results if the MERRA-2 biases changed systematically over time.

Essentially, the sampling correction replaces known larger biases due to OSIRIS sampling issues with potential smaller biases from MERRA-2. The end result is an improved dataset, as shown in Sect. 2.2 and Appendix B.

In the tropical lower stratosphere, ozone variability is minimal (see e.g., Fig. A1c). The standard deviations increase (by a few percent) likely because MERRA2 has slightly more variability in that region, and the correction transfers this variability to the OSIRIS dataset. On the other hand, Fig. 1a shows the relative change in standard deviations, and since the standard deviations themselves are small for both the corrected and uncorrected data, the relative differences are not necessarily the best metric for the tropical lower stratosphere.

L127: "Multiplying each the OSIRIS profile with …"

"Multiplying each of the OSIRIS profiles with …"

Removed 'the' from the sentence.

L129: "The correction does not attempt to remove longitudinal variability (the dominant variability in the MZM), which is well sampled by OSIRIS."

I think it's worth mentioning that nor does it remove any of the random variance that is naturally present in the data. This sounds vaguely familiar except you're using model data for the correction instead of the regression fit itself. I suppose that makes sense since your regression is being applied to each latitude bin separately and thus you don't have a measure of the seasonally varying spatial gradient.

Added another sentence on L130 to say that "Random variability in the data is not removed either.".

L132: "The method includes a simple diurnal correction as well, since the local time of the MERRA-2 reference profiles is fixed to noon, whereas average descending node local times for OSIRIS are between 6–8 am. It should be noted that the ozone diurnal cycle is not represented in MERRA-2 before 2004, i.e. before day and night measurements from MLS were available for assimilation (Wargan et al., 2017)."

Can you elaborate more on how the diurnal correction is derived / applied? Also, if the diurnal cycle is not represented in MERRA-2 prior to 2004, what do you do with the OSIRIS data during this time period?

The diurnal correction is implicit, since we use 3 hour MERRA-2 data for the correction, and each profile is corrected to noon. We use the same correction methodology regardless of the date, since the MERRA-2 dataset does not differentiate between the time periods and we don't model any diurnal differences. The effect on the data is likely small, as seasonal and latitudinal changes dominate. We included this section simply to point out an additional source of uncertainty in the corrected dataset.

Replaced "simple" with "implicit" in the sentence at L132.

L161: "If multiple tropopauses are present, the second tropopause …"

I think we all know this, but it's probably worth stating somewhere that the second tropopause is higher than the first. Maybe simply "… the second, (i.e., higher) tropopause …"

This part now reads: "[…] the second (i.e., the higher) tropopause is used [...]"

L161: "data up to and including the altitude level that contains the tropopause are discarded."

At least for SAGE data, we usually recommend excluding data up to and including 1 km above the tropopause altitude since that is roughly the vertical resolution of the data.

We decided to keep the original filtering approach in order to process all three instruments consistently. With the original method, the first data point in each profile is 0.5-1.5 km above the tropopause altitude. In addition, the MLR and DLM fits are only shown for altitude bins that are fully above the mean + 1 sigma tropopause altitude for the given latitude (L175-176). This ensures that potential tropospheric contamination is reduced while preserving enough data for meaningful lower stratospheric fits. A more restrictive filter would also limit the usefulness of the tropopause-relative (TR) dataset, since the first TR data point is already at 1.5 km using the current approach.

L168: "SAGE II and SAGE III/ISS data are then bias-corrected using OSIRIS as the baseline, and the deseasonalized anomalies are averaged."

I assume this is done by adjusting their mean values to be the same in the overlap period? If the regressions are performed on anomalies, why notAlso, as a general recommendation on wording choice, I would avoid using the word "bias" when possible (i.e., perhaps use "offset" instead) as the connotation around the word "bias" is that one instrument is inherently wrong.

The relevant sentence now reads: "SAGE II and SAGE III/ISS data are then adjusted such that the mean difference with respect to OSIRIS in the overlap period is zero, […]". Unfortunately the first part of the question appears to have been cut off when the comment was submitted.

L184: "The GloSSAC data (v2.1, 1979-2020) is extended to 2021 by extrapolating the last value."

Perhaps state that GloSSAC is extended "through" 2021 (i.e., December 2021) as the current wording could be misconstrued to mean just up to 2021 (i.e., December 2020). Normally I wouldn't advise simply extending the aerosol data set using the last value considering how things have been recently, but 2021 was a comparatively benign year (for future work and reference, 2022 is not a benign year). Also just as an FYI, there is a v2.2 that adds 2021 (https://asdc.larc.nasa.gov/project/GloSSAC).

Clarified the sentence to say that the data is extended to the end of 2021. The results still use the v2.1 data, since initially the v2.2 GloSSAC dataset was not available. The data link only went live close to the completion of these revisions, after all the additional DLM fits were completed using the original v2.1 dataset.

L211: "The weighted MLR trend results, however, are sensitive to the exact correction method chosen …"

And how the data from the different instruments are merged together. The LOTUS Report specifically mentioned how using a weighted MLR was almost impossible with a pre-merged data set as the heteroscedasticity correction would ideally need to be applied to each individual instrument's data set separately.

We tested both approaches (correction pre- or post-merge), and this sentence was meant to summarize those results. We added a note in parentheses on L212 to say that the method chosen affects the results

"even when the correction is applied prior to merging". While we can perform the correction prior to merging, development of an effective heteroscedasticity correction method is beyond the scope of this paper.

L246: "The variable turnaround dates highlight one disadvantage of MLR, where the turnaround period is a fixed parameter that leads to endpoint anomalies in the trend results"

This is somewhat dependent on the trend model as the ILT method helps to mitigate this effect by generally avoiding fitting a trend line near the potential turnaround and the EESC EOFs don't specify a turnaround time at all (though they are constrained to the range of potential turnaround times governed by mean age of air).

While the magnitude of the effect does depend on the trend model, the endpoint anomalies are there regardless. Changing the ILT period still has a large effect on the results, as different periods are fitted for the ozone decrease and increase. The DLM fit on the other hand makes no assumptions about the ozone behavior near the turnaround point. We only discuss the piece wise and ILT MLR fits in the manuscript, and so did not add a reference to EESC fits (as those models rely on a different set of assumptions).

L252: "… but the negative changes indicate a pause in ozone recovery nonetheless."

Is there any physical reason to believe in this second turnaround in the NH mid-latitude upper stratosphere?

See also the response to the last general comment. We can only speculate (as we do in the manuscript) about what might cause such a turnaround. We do point out in the first half of the sentence on L251-252 that the ozone decrease from the turnaround to 2021 is not statistically significant, which means that the turnaround itself isn't either.

L278: "To test the importance of the selected end year, we recalculated the SOS dataset and the corresponding DLM fit with data ending in 2017–2021."

Please add "not shown" to indicate there isn't any figure associated with these runs. Also, please modify the wording to be more explicit that you reran the DLM with separate runs each ending at a different year in the range of 2017 to 2021.

The relevant section now reads:

"To evaluate the importance of the selected end year, we tested the years 2017-2021. For each of the five end years, we recalculated the SOS dataset and reran the DLM fit (results not shown). Using a different end year changes the relative anomalies slightly, as the dataset available for the OSIRIS and SAGE III/ISS seasonality calculation is truncated."

L312: "… and likely stronger than SOS data show."

Not necessarily, that would depend on the drift of each instrument.

Replaced 'likely' with 'possibly', in line with the response to the general comment on SOS – MLS differences. Made the same change on the conclusions (L406) as well.

L332: "Trend variability is low, however: using twice the standard deviation of the 2017–2021 changes as the significance threshold, the region of significant trends approximately matches the 90 % significance contour in Fig. 2b."

I had to read this sentence a few times. I wonder if there's a more explicit/descriptive phrasing that could be used instead of this most efficient phrasing.

Rephrased this (and the preceding) sentence to say that

"At mid-latitudes, trend patterns remain largely unchanged, although the confidence in negative changes decreases with earlier end years. Overall, trend variability as a function of end year is small compared to the trend magnitudes.".

L348: "that dataset is substantially different from the SOS composite used here."

Might reiterate it is because only the SOS composite attempts to apply sampling corrections.

Added a note in parentheses to the end of the sentence to clarify the differences: "different instruments and data versions, no OSIRIS sampling correction"

L361: "the original DLM trends"

Perhaps add "shown in Fig. 2" as a reminder

Added a reference to Fig. 2b.

L361: "… interpolated to approximate TR coordinates using the mean height of the first tropopause in the SOS dataset."

I question how robust this is as a method since the tropopause height has seasonality, though perhaps its impact is only felt strongly near the tropopause.

The method is intended only to allow easier visual comparison on the same TR grid. Since interpolation is done on the actual trend values (i.e., the calculated 2000-2021 change), the seasonality of tropopause heights should not impact the results.

L363: "The results are similar above 8–10 km, indicating that TR coordinates have little impact in the middle stratosphere and above."

They are similar where gradients in the trend are small, otherwise I wouldn't say they're that similar.

Since the discussion focuses on the changes just above the tropopause, we removed the reference to the middle stratosphere by replacing this sentence and the next one with "The largest differences appear below 8-10 km, as expected."

L370: "TR trends remain mostly not significant in the tropics for dataset end years of 2017–2021"

Again, perhaps add "not shown".

Made the change.

L424: "The positive offset in the upper stratosphere is most likely caused by the updated ozone cross-section, while in the lower stratosphere the offset is mainly the result of the updated aerosol retrieval (Sect. 2.1)."

Shouldn't cross-section changes impact uniformly everywhere? If so, why the negative change in middle stratosphere?

The OSIRIS measurement vector is constructed from wavelength pairs/triplets that cover different (overlapping) vertical ranges (Degenstein et al., 2009, DOI:10.5194/acp-9-6521-2009), and so wavelength-dependent changes between the cross-sections can lead to altitude-dependent differences between the retrieval results. The different temperature dependence of the cross-sections might play a role as well. We cannot fully separate the effects of individual changes to the retrieval algorithm, as the OSIRIS data version increments (between 5.10 and 7.2) contain multiple changes each.

L452: "Here we include a seasonal term with annual and semiannual components …"

Is there still some constraint on this or is it incorporated like a standard MLR proxy? I only ask because if there is some constraint / prior, the sampling-induced differences may not adhere to this.

The seasonal term is introduced as an additional DLM component, not as regressors. The seasonal fit is allowed to vary dynamically, and, as for the trend, only the prior on the allowed time-wise variability in the seasonal components is specified (see e.g., Laine et al., 2014; DOI: 10.5194/acp-14-9707-2014). We modified the relevant sentence to say that "we include a dynamic seasonal term […]".

Fig. 2 Caption: "represents lack of statistical significance"

Do you mean at the 95% CI?

Specified that the significance level is 2 sigma for the MLR results (see response to RC1).

The contour lines drawn in several of the figures are almost impossible to distinguish with zooming in extensively. Please modify the graphic to increase the spacing between dots/dashes so that the reader can tell which is which when viewed at normal resolution.

Made all contour lines slightly thicker and increased the spacing for the dotted and dashed lines on the DLM plots.

Since the trend plots do not have data contour lines on them (i.e., they have significance contour lines), it might be helpful to set a specific color increment (e.g., something like 0.5–1.0% per color transition). The continuous color gradation makes it very difficult to have a sense of what the numbers might actually be.

All colormap figures that did not have value contours (Figs. 2, 3, 5a, 6, A2a, and B1) are now plotted using discrete color scales with 1% increments. Figures that did have value contours are still plotted using the original continuous color scale.

---

## Author Response (AR2)

**Response to Referee Comments**

Suggested minor changes by Referee 1:

line 79: "To improve the trend results" -> "To improve the used monthly zonal mean values"

    Changed the relevant sentence to: "To improve the monthly zonal mean time series […]".

line 82: "recovery largely stopped" -> "recovery appears to pause" or "recovery is pausing"

    Changed the relevant phrase to "recovery appears to pause".